# Glycolic acid and D-lactate—putative products of DJ-1—restore neurodegeneration in FUS - and SOD1-ALS

Arun Pal[1,2], Dajana Grossmann[3], Hannes Glaß[3], Vitaly Zimyanin[1,4,5], René Günther[1,6], Marica Catinozzi[9], Tobias M Boeckers[7], Jared Sterneckert[8], Erik Storkebaum[9], Susanne Petri[10], Florian Wegner[10], Stephan W Grill[11,12], Francisco Pan-Montojo[13], Andreas Hermann[3,14,15]

**Amyotrophic lateral sclerosis (ALS) leads to death within 2–5 yr. Currently, available drugs only slightly prolong survival. We present novel insights into the pathophysiology of *Superoxide Dismutase 1* (SOD1)- and in particular *Fused In Sarcoma* (FUS)-ALS by revealing a supposedly central role of glycolic acid (GA) and D-lactic acid (DL)—both putative products of the Parkinson's disease associated glyoxylase DJ-1. Combined, not single, treatment with GA/DL restored axonal organelle phenotypes of mitochondria and lysosomes in FUS- and SOD1-ALS patient-derived motoneurons (MNs). This was not only accompanied by restoration of mitochondrial membrane potential but even dependent on it. Despite presenting an axonal transport deficiency as well, TDP43 patient-derived MNs did not share mitochondrial depolarization and did not respond to GA/DL treatment. GA and DL also restored cytoplasmic mislocalization of FUS and FUS recruitment to DNA damage sites, recently reported being upstream of the mitochondrial phenotypes in FUS-ALS. Whereas these data point towards the necessity of individualized (gene-) specific therapy stratification, it also suggests common therapeutic targets across different neurodegenerative diseases characterized by mitochondrial depolarization.**

# Introduction

Amyotrophic lateral sclerosis (ALS) is the most common motor neuron disease, with an estimate of 17,000 patients and ~5,000 new cases annually only in Europe (Marin et al, 2017). Worldwide incidence is ~1.6 cases per 100,000 persons annually (Brotman et al, 2023). Compared with other neurodegenerative disorders, ALS exhibits the fastest fatality rate, with an expected survival time of 2–5 yr (Brooks et al, 2000; Brotman et al, 2023). Until today, ALS is an incurable disease with Riluzole, Sodium Phenylbutyrate/ Taurursodiol, and Edaravone being the only approved and commercially available putative disease modifying treatments (Edaravone and Sodium Phenylbutyrate/Taurursodiol not in the EU) (Chen, 2020; Kiernan et al, 2021). However, these treatments show limited efficacy (Bensimon et al, 1994; Lacomblez et al, 1996a, 1996b; Chen et al, 2016). For example, the impacts of Edaravone including real-world settings are mixed, with some studies showing no effects (Witzel et al, 2022; Writing Group & Edaravone MCI-186 ALS 19 Study Group, 2017; Abraham et al, 2019; Lunetta et al, 2020). Overall, Riluzole can be expected to delay time to death or time to tracheostomy for patients with ALS by about 3 mo (Hinchcliffe & Smith, 2017).

ALS appears in familial (~10% of cases) and sporadic forms (~90% of cases) (Turner et al, 2017) and is caused by the degeneration of MNs in the spinal cord and brain stem (lower MNs) and the motor cortex (upper MNs), progressively resulting in paralysis and death (Zarei et al, 2015). Mutations in the genes *C9ORF72*, *SOD1*, *FUS*, and *TARDBP* are the most frequent monogenetic forms associated with familial ALS (Hou et al, 2016; Müller et al, 2018). The pathogenic mechanisms underlying MN death have been extensively studied. Increased glutamate signaling and intracellular calcium levels (excitotoxicity), ER stress, mitochondrial dysfunction, oxidative stress because of the increase in reactive oxygen species (ROS), dysregulated transcription and RNA processing, protein misfolding

[1]Division for Neurodegenerative Diseases, Department of Neurology, Technische Universität Dresden, Dresden, Germany   [2]Dresden High Magnetic Field Laboratory (HLD-EMFL), Helmholtz-Zentrum Dresden-Rossendorf (HZDR), Dresden, Germany   [3]Translational Neurodegeneration Section "Albrecht Kossel", Department of Neurology, University Medical Center Rostock, University of Rostock, Rostock, Germany   [4]Department of Molecular Physiology and Biological Physics, University of Virginia, School of Medicine, Charlottesville, VA, USA   [5]Center for Membrane and Cell Physiology, University of Virginia, School of Medicine, Charlottesville, VA, USA   [6]Deutsches Zentrum für Neurodegenerative Erkrankungen (DZNE), Dresden, Germany   [7]Institute for Anatomy and Cell Biology, Ulm University, as well as Deutsches Zentrum für Neurodegenerative Erkrankungen, Ulm, Germany   [8]Center for Regenerative Therapies Dresden, Technische Universität Dresden as well as Medical Faculty Carl Gustav Carus of TU Dresden, Dresden, Germany   [9]Donders Institute for Brain, Cognition and Behaviour and Faculty of Science, Radboud University, Nijmegen, Netherlands   [10]Department of Neurology, Hannover Medical School, Hannover, Germany   [11]Max Planck Institute of Molecular Cell Biology and Genetics, Dresden, Germany   [12]Cluster of Excellence Physics of Life, Technische Universität Dresden, Dresden, Germany   [13]Department of Psychiatrie and Psychotherapy, LMU University Hospital, LMU Munich, Munich, Germany   [14]Deutsches Zentrum für Neurodegenerative Erkrankungen (DZNE) Rostock/Greifswald, Rostock, Germany   [15]Center for Transdisciplinary Neurosciences Rostock (CTNR), University Medical Center Rostock, University of Rostock, Rostock, Germany

Correspondence: Andreas.Hermann@med.uni-rostock.de
Francisco Pan-Montojo's present address is Neurological Clinic am Sorpesee, Sundern-Langscheid, Germany

and aggregation, dysregulated endosomal trafficking, impaired axonal transport, and neuroinflammation are key components involved in the pathogenesis of ALS (for review, see Tadic et al [2014] and Weishaupt et al [2016]). Nevertheless, the common putative devastating pathophysiological cascade as a whole remains to be understood. However, a common early sign of degeneration is a dying back of the neurons with early axonal trafficking deficits in many genetic ALS forms (Bilsland et al, 2010; Moller et al, 2017; Kreiter et al, 2018; Naumann et al, 2018; Pal et al, 2018, 2021). The particular vulnerability of MNs compared with other neuronal groups is still a matter of debate; however, it might involve both high expressions of AMPA receptors that lack the calcium-impermeable GluR2 subunit, which makes them more prone to excitotoxicity and imbalances in intracellular $Ca^{2+}$ homeostasis (Williams et al, 1997). Moreover, MNs are known to express low levels of $Ca^{2+}$-buffering proteins, which increases their vulnerability (Ince et al, 1993). In addition, MNs strongly rely on optimal mitochondrial function, because of their high metabolic demands, and are therefore more prone to cell death when mitochondrial activity is dysregulated. Overall, the crosstalk between $Ca^{2+}$, the endoplasmic reticulum (ER), and mitochondrial function as well as oxidative stress seems to be crucial in the development of ALS pathology (Tadic et al, 2014).

Furthermore, we recently showed mitochondrial depolarization as early events in SOD1- and in particular FUS-ALS patients-derived MNs (Naumann et al, 2018; Gunther et al, 2022). Most importantly, restoration of mitochondrial inner membrane potential delayed neurodegeneration in FUS-ALS (Naumann et al, 2018). ALS-causing mutations in *FUS* are mainly localized in its nuclear localization sequence (NLS) and thus cause a cytoplasmic mislocalization (Japtok et al, 2015; Szewczyk et al, 2021, 2023). This is accompanied by a loss of nuclear FUS function including a lack of proper recruitment of FUS to DNA damage sites (DDS) and DNA damage repair (DDR) (Naumann et al, 2018; Wang et al, 2018), a mechanism downstream of poly(ADP-ribose) polymerase 1 (PARP1) (Mastrocola et al, 2013; Rulten et al, 2014). Of note, perturbation of DDR was always associated with distal axonal organelle impairments: inhibition of FUS recruitment to DDSs in the WT along with its cytosolic aggregation led to distal axonal organelle motility impairment (lysosomes and mitochondria) and loss of mitochondrial inner membrane potential (detailed in Naumann et al [2018] and Pal et al [2018]). In addition, we found that FUS-mediated DDR in the nucleus was upstream of the distal axonal organelle phenotypes, as pharmacological inhibiton of mitochondrial function and motility directly at the distal axon compartment had no impact on the DDR.

Having shown mitochondrial depolarization without dominant affection of mitochondrial respiration, we were looking for candidates, which might be able to restore these phenotypes in iPSC-derived MNs. To this end, we came across glycolic acid (GA) and D-lactate (DL). Glycolic acid (GA) and D-lactate (DL) both occur naturally in the cell as products of DJ-1 (Lee et al, 2012). Most importantly, however, loss of DJ-1 leads to a drastic inner mitochondrial membrane depolarization, which could be restored by supplementing GA and/or DL (Toyoda et al, 2014), making them ideal candidates for restoration of above-mentioned SOD1- and particularly FUS-ALS-induced inner mitochondrial membrane depolarization phenotypes. DJ-1 converts the reactive aldehydes

glyoxal and methylglyoxal to GA and DL, respectively (Thornalley, 2003; Lee et al, 2012). The protein glyoxylase DJ-1, encoded by the *PARK7* gene, is known as a redox-dependent chaperone with neuroprotective potential. Loss-of function mutations cause early-onset autosomal recessive PD (Hague et al, 2003). DJ-1 overexpression protects dopaminergic neurons against PD, whereas DJ-1 deficiency leads to profound loss of dopaminergic neurons (Kim et al, 2005). DJ-1 is integral for maintaining mitochondrial potential, $Ca^{2+}$ homeostasis and ATP production. Of note, loss of DJ-1 did not affect mitochondrial respiration but increased ROS production and mitochondrial permeability transition pore opening (Giaime et al, 2012), phenotypes which we recently identified also in FUS-ALS MNs (Zimyanin et al, 2023). GA can support the mitochondrial membrane potential and neuronal survival (Toyoda et al, 2014), improve mitochondrial energy production, thereby increasing the levels of NAD(P)H (Bour et al, 2021 *Preprint*) and can also reduce oxidative stress via a glutathione-mediated pathway (Diez et al, 2021). Therefore, we speculated that GA and DL might be a treatment option for MNs differentiated from induced pluripotent stem (iPS) cells derived from fibroblasts of ALS patients with different ALS-causing mutations and compared it with the standard of care treatment (Riluzole).

# Results

### GA and DL restore axonal trafficking in FUS-ALS mutants

Products of the PD-related glyoxalase DJ-1, namely GA and DL, were reported to support mitochondrial membrane potential and neuronal survival in PD animal models (Toyoda et al, 2014). We have recently shown significant mitochondrial dysfunction in FUS-ALS patient-derived MNs, including severe axonal trafficking deficiency, mitochondrial fragmentation (i.e., less elongated) and loss of mitochondrial membrane potential (Naumann et al, 2018; Pal et al, 2018). In particular, the latter was rescued by GA and DL in the PD models (Toyoda et al, 2014). Thus, we investigated whether the treatment of GA and DL together is able to rescue FUS-ALS MN phenotypes as well. To this end, patient-derived spinal MNs (Table 1 for details on mutations and patients) were matured for 21 d in vitro (DIV) in microfluidic chambers (MFCs)—time points at which neurons were shown to exhibit typical neuronal electrophysiological properties (Naujock et al, 2016; Bursch et al, 2019), but importantly at which mutant cells exhibited axon trafficking and mitochondrial phenotypes (Naumann et al, 2018; Pal et al, 2018)—and then treated for 24 h at both sites (distal and proximal) with GA and DL (each 1 mM) and imaged using Mitotracker JC-1 and Lysotracker (Fig 1). The combination of GA and DL did completely restore axon trafficking phenotypes of mitochondria and lysosomes of FUS-ALS spinal MNs (Figs 1A–C and S1 for individual cell lines). Remarkably, the enantiomer of DL, L-lactate (LL), in combination with GA at similar concentrations did not lead to any alteration of these axonal trafficking phenotypes in mutant FUS-ALS MNs (Fig S2A–C). Moreover, treatment with either GA, DL, or LL alone had no effect as well (Figs 1A, S2A–C, and S3A). Specifically, whereas up to 20 mM of GA or DL alone had no effect on axon trafficking, the $EC_{50}$ of the

**Table 1. Patient/proband characteristics.**

| Genotype | Cell line | Sex | Age at biopsy | Mutation | Primarily characterized in |
|---|---|---|---|---|---|
| Wt | Ctrl1 | Male | 48 | — | Japtok et al (2015) |
| Wt | Ctrl2 | Female | 43 | — | Reinhardt et al (2013) |
| Wt | Ctrl3 | Female | 48 | — | Reinhardt et al (2013) |
| IGC | FUS-WT eGFP[het] | Isogenic to FUS R521C and FUS-P525L GFP | N/A | — | Naumann et al (2018) |
| IGC | SOD1 D90A igc | Isogenic to SOD1 D90A | N/A | — | Bursch et al (2019) |
| Mt | TDP43 S393L[het] | Female | 87 | S393L | Kreiter et al (2018) |
| Mt | TDP43 G294V[het] | Male | 46 | G294V | Kreiter et al (2018) |
| Mt | SOD1 D90A[hom] | Female | 46 | D90A | Naujock et al (2016) |
| Mt | SOD1 A4V[het] | Female | 73 | A4V | Gunther et al (2022) |
| Mt | SOD1 R115G[het] | Male | 59 | R115G | Naujock et al (2016) |
| Mt | FUS R521C[het] | Female | 58 | R521C | Naumann et al (2018) |
| Mt | FUS R521L[het] | Female | 65 | R521L | Naumann et al (2018) |
| Mt | FUS R495X[het] | Male | 29 | R495QfsX527 | Naumann et al (2018) |
| Mt | FUS-P525L eGFP[het] | Isogenic to FUS R521C and FUS-WT GFP | N/A | P525L | Naumann et al (2018) |
| Wt | HeLa BAC FUS-eGFP WT | Female | N/A | — | Poser et al (2008) and Maharana et al (2018) |
| Mt | HeLa BAC FUS-eGFP P525L | Female | N/A | P525L | Poser et al (2008) and Maharana et al (2018) |

combination of GA and DL was 485 $\mu$M each for restoration of axonal trafficking (Fig S3B). Furthermore, the combinatorial treatment with 1 mM GA and DL restored mitochondrial fragmentation (Fig 1D) and mitochondrial inner membrane potential (Fig 1E) whereas the combinatorial treatment with 1 mM GA and LL had no effect on mitochondrial fragmentation either (Fig S2D).

### GA and DL restore FUS nuclear cytoplasmic mislocalization and recruitment to nuclear laser-irradiated DNA damage sites

ALS-causing mutations in *FUS* are mainly localized in its NLS and thus causing a cytoplasmic mislocalization, i.e., aggregation (Japtok et al, 2015; Szewczyk et al, 2021, 2023). This is accompanied by a loss of nuclear FUS function including a lack of proper recruitment of FUS to DNA damage sites and DNA damage repair (DDR) (Naumann et al, 2018; Wang et al, 2018). Our previous studies always showed a dependency of proper mitochondrial membrane potential and trafficking on proper DNA damage repair, and rescuing DNA damage repair always restored the perturbed mitochondrial membrane potential and trafficking in distal axon parts, but not vice versa. Thus, we wondered whether GA and DL treatment also affects this nuclear function of FUS. First, we investigated whether treatment with GA and DL influenced FUS nuclear/cytoplasmic distribution. To this end, we used engineered HeLa cells expressing a bacterial artificial chromosome of either WT or P525L mutant FUS tagged with eGFP at the carboxyl-terminus (Poser et al, 2008). Similar to axonal trafficking, individual treatment of the P525L mutant with either GA or DL did not influence the count of cytoplasmic FUS aggregates whereas co-treatment with both led to a full rescue back to the WT

level (Fig 2A and B). We next used targeted laser irradiation to induce DNA strand breaks at defined nuclear positions as described previously (Naumann et al, 2018). Whereas mutant FUS cells showed complete loss of FUS recruitment in Mock-treated conditions, this phenotype was restored in case of combined GA and DL treatment but not by single treatments with GA or DL (Fig 2C and D). The rescue effect of GA and DL was observed at a similar $EC_{50}$ compared with restoration of axonal trafficking (496.5 $\mu$M, Fig S4A and B) with a remaining minor delay compared with the WT control (Figs 2D and S4A). Regarding the enantiomer LL, we found that the combination of GA and LL or LL alone at similar concentrations was incapable to rescue the cytosolic aggregation of FUS (Fig S5A and B) as well as the failed recruitment to laser-irradiated DNA damage sites (Fig S5C and D) in the mutant, consistent with the axonal trafficking phenotypes (Fig S2). We finally used CRISPR/CAS9 gene-edited iPSC-derived spinal MNs (Table 1) either expressing WT or P525L FUS-eGFP and proved that 10 mM co-treatment with GA and DL was also able to restore FUS recruitment to laser-irradiated DNA damage sites in spinal MNs (Fig 2E and F) as well as the cytosolic mislocalization of FUS in the mutant (Fig S6A and B), thereby validating our findings in the HeLa ALS-FUS model (Fig 2A–D).

### Axon trafficking but not FUS DNA damage site recruitment depends primarily on proper mitochondrial function

Both axon trafficking and DNA damage repair are highly energy demanding and thus depend on the availability of energy in the cell (Martire et al, 2015; Naumann et al, 2018). Postmitotic neurons rely

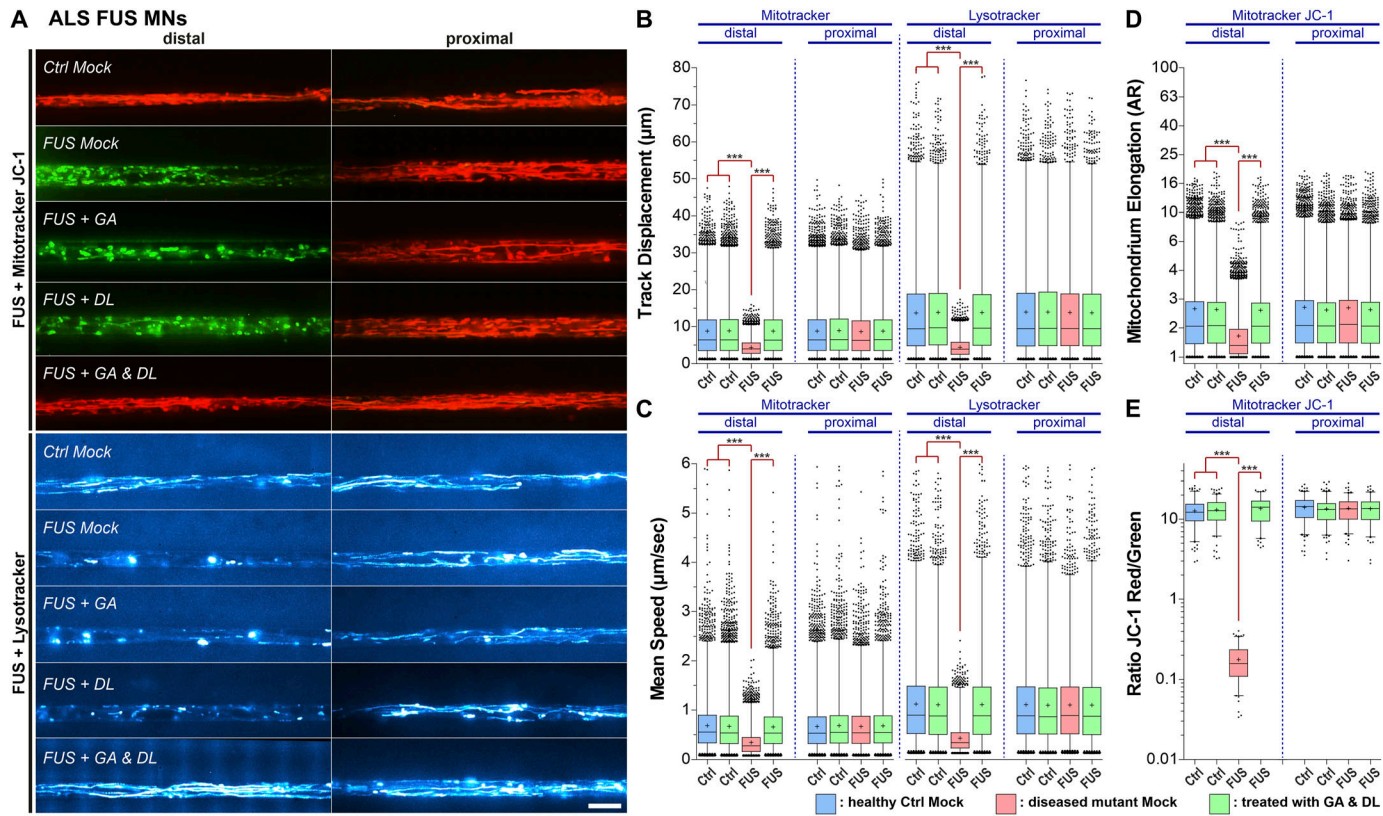

**Figure 1. GA and DL together rescue axonal trafficking defects in amyotrophic lateral sclerosis-FUS mutants.**

Patient-derived spinal MNs were matured for 21 d in vitro in microfluidic chambers, double treated for 24 h at both sites (distal and proximal) with GA and DL (each 1 mM) and imaged live at the distal (left) versus proximal (right) channel end with Mitotracker JC-1 (red/green) and Lysotracker (cyan hot). **(A)** Maximum intensity projections of videos visualize organelle moving tracks in axons. Shown are single, representative microchannels of the microfluidic chamber microgroove barrier either at the distal (left) or proximal (right) end, which were inhabited by a protruding bundle of typically 5–20 axons. Processive motility results in straight, longer trajectories, whereas immobile organelles project as punctae. Representative examples from the mutant FUS (Fig S1, Table 1) and control (Ctrl) line pools are shown as follows: FUS: FUS R521C, Ctrl: Ctrl1. Note the exclusively distal loss of lysosomal and mitochondrial motility and its inner membrane potential (JC-1 green) in FUS Mock compared with Ctrl (JC-1 red), which were both rescued through GA and DL double treatment but not through GA or DL alone even at 20 mM (Video 1 and Video 2). Scale bar = 10 μm. **(B, C, D, E)** Box plots quantifications of various tracking and morphology parameters deduced from videos from (A) as per organelle values (i.e., each data point presents one individual organelle) for the mutant FUS and Ctrl cell line pool, except of (E) showing mean values per image. For mutant FUS, data from the FUS R521C, R521L, R495X, and FUS P525L-eGFP lines were pooled (Table 1). For WT Ctrl, data from the Ctrl1, Ctrl2, Ctrl3, FUS WT-GFP, and SOD1 D90A igc lines were pooled (Table 1). For individual cell lines, refer to Fig S1. Box: 25–75% interquartile range, horizontal line: median, cross: mean, whiskers: non-outlier range (99% of data), dots outside whiskers: outliers, Ctrl Mock is shown in pale blue, diseased mutant mock in pale red, double treatment with GA and DL in pale green. **(B, C, D, E)** A one-way ANOVA with either a Kruskal-Wallis post hoc test to account for the non-normal, top-tailed data distributions (B, C, D) or Bonferroni post hoc test for the normal distributions (E) was used to reveal significant differences in pairwise comparisons. Asterisks: highly significant alteration in pairwise comparisons as highlighted by brown brackets above data, ***$P ≤ 0.001$, all other pairwise comparisons were not significantly different. **(B, C)** Note the drastic reduction in exclusively distal track displacement (B) and mean speed (C) in FUS Mock that was rescued through GA and DL double treatment. **(D)** Note the drastic reduction in exclusively distal mitochondria elongation (fragmentation) in FUS that was rescued by GA and DL double treatment. **(E)** Note the loss of exclusively distal mitochondrial inner membrane potential in FUS that was rescued by GA and DL double treatment.

mainly on oxidative phosphorylation (OXPHOS) to generate ATP (Zheng et al, 2016). We thus interfered with mitochondrial OXPHOS in WT control MNs to test whether this inhibition is already sufficient to induce axonal trafficking deficiency as well as lack of poly(ADP)ribose-dependent FUS recruitment to DNA damage sites. WT MNs (Ctrl 1–3, Table 1) were treated with 10 μM Oligomycin A, an inhibitor of the respiratory chain complex V, or with 10 μM of the uncoupler carbonyl cyanide 3-chlorophenylhydrazone (CCCP) for 24 h on the distal site of MFCs. The resulting dysfunction of axonal mitochondria induced phenocopies of FUS-ALS axonal trafficking defects (Fig 3A). Conversely, the same treatments did not interfere with FUS-recruitment to laser-induced DNA damage sites in MNs expressing WT FUS-eGFP (Table 1) (Fig 3B and C), albeit an increase

in cytosolic FUS (Fig 3D) that was insufficient to interfere with its nuclear function.

## GA and DL restore nuclear phenotypes by restoring NAD metabolism

We recently reported that restoration of nuclear phenotypes in FUS-ALS MNs subsequently led to a restoration of axonal trafficking and disappearance of cytosolic FUS aggregates (Naumann et al, 2018; Pal et al, 2018). Therefore, we were wondering about the mechanisms by which GA and DL might restore nuclear functions in FUS-ALS including proper recruitment of FUS to sites of DNA damage. FUS is recruited to DNA damage sites downstream of

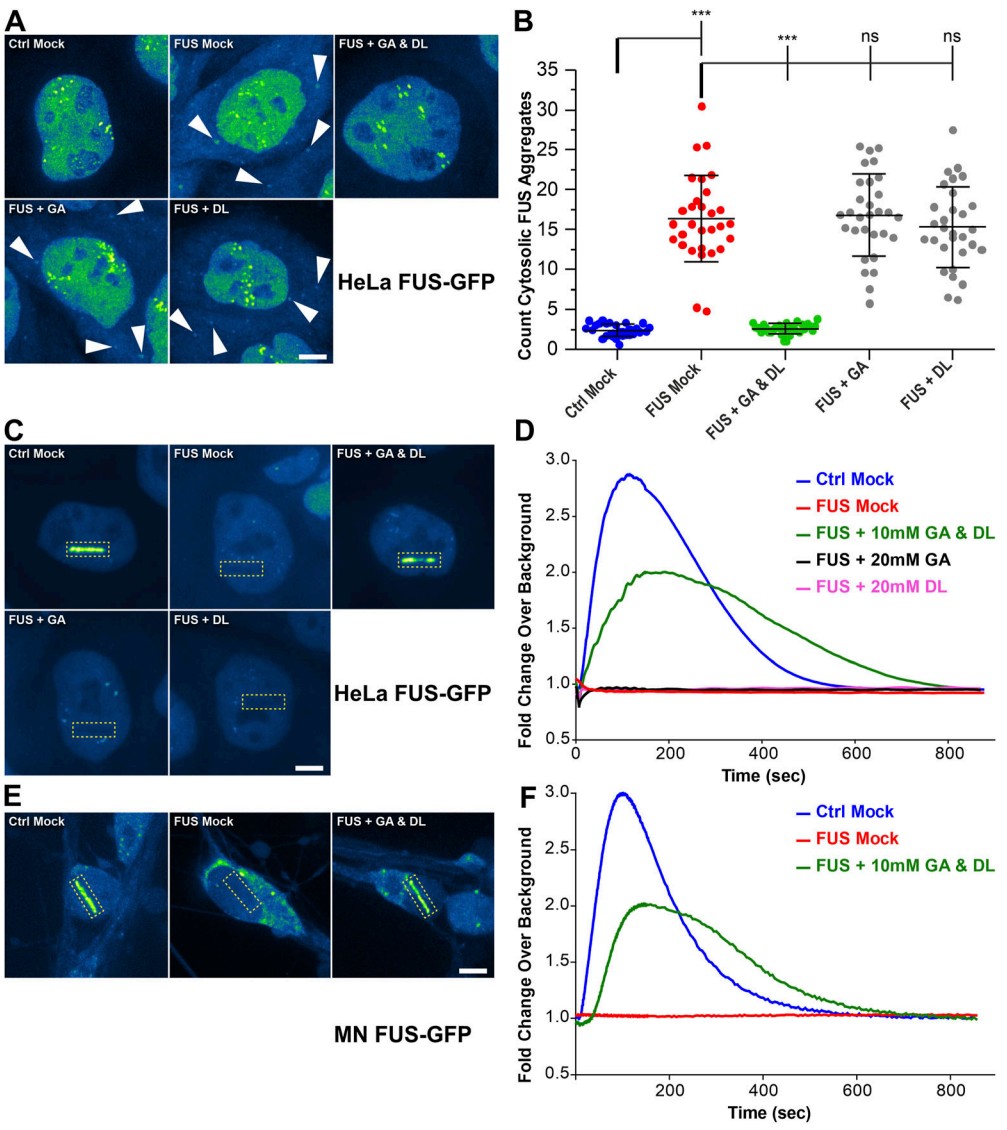

**Figure 2. GA and DL together rescue recruitment of FUS to nuclear DNA damage sites in mutant amyotrophic lateral sclerosis FUS cells.**
**(A)** GA and DL double treatment rescues from cytosolic FUS aggregation. Shown are maximum intensity projections of confocal Z-stacks imaged live in the transgenic bacterial artificial chromosome HeLa cell model expressing normal FUS-eGFP WT (Ctrl) or mutant FUS-eGFP P525L (FUS) (Table 1). The eGFP intensity is shown in the LUT "Green Fire Blue" of the FIJI software, i.e., low eGFP intensities are shown in blue and high intensities in green shades, no nuclear HOECHST staining or alike was used. Each viewing field centres a single nucleus with its surrounding cytosol. The cytosol particularly in Ctrl cells appears very dark similar to intercellular blank background because of its low content of eFUS-GFP. Note the occurrence of cytosolic FUS aggregates in FUS Mock as compared with Ctrl Mock (arrowheads) that were rescued through GA and DL double treatment for 24 h (each 10 mM) but not through GA or DL alone even at 20 mM. Scale bar = 10 µm.
**(B)** Quantification of (A) as counts of cytosolic FUS aggregates per cell, N = 30 images, each data point of the scatter dot plots presents one mean value per image, whiskers show the STDEV, centre lines the median. Note the drastic increase in FUS Mock (red dots) as compared with Ctrl Mock (blue dots) and its reversion back to Ctrl levels through GA and DL double treatment (green dots) but not through GA or DL alone (grey dots). A one-way ANOVA with Bonferroni post hoc test was used for the normal distributions of the data sets to reveal significant differences in pairwise comparisons as highlighted by brackets above data. Asterisks: highly significant alteration in indicated pairwise comparison, ***$P ≤ 0.001$ and ns: no

significant difference. **(C)** Transgenic bacterial artificial chromosome HeLa cells from (A) were double treated for 24 h with GA and DL (each 10 mM). Recruitment-withdrawal of FUS-GFP to UV laser cuts in nuclei (boxed area) was then imaged live (Video 3). Shown are single video frames at 150 s when the eGFP intensity was around its maximum. The eGFP intensity is shown in the LUT "Green Fire Blue" of the FIJI software, i.e., low eGFP intensities are shown in blue and high intensities in green shades, no nuclear HOECHST staining or alike was used. Each viewing field centres a single nucleus with its surrounding cytosol. The cytosol particularly in Ctrl cells appears very dark similar to intercellular blank background because of its low content of FUS-eGFP. Note the failed recruitment in FUS Mock as compared with Ctrl Mock and its rescue through GA and DL double treatment but not through GA or DL alone even at 20 mM. Furthermore, in case of failed FUS-eGFP recruitment (i.e., FUS Mock, FUS + GA, FUS + DL), the laser beam left a dark line because of photo bleaching that is not to be mistaken for FUS-eGFP withdrawal from the DNA damage site. Scale bar = 10 µm.
**(D)** Quantification of (C). Note the failed recruitment in FUS Mock (red curve) over the entire recording time of 850 s whereas GA and DL double treatment (green curve) fairly restored the recruitment-withdrawal towards Ctrl kinetics (blue curve) whereas neither GA nor DL alone (black and pink curve, respectively) rescued the FUS-eGFP recruitment. **(E)** Patient-derived isogenic spinal MNs expressing normal (Ctrl) or mutant P525L (FUS) FUS-eGFP (Table 1) were matured for 21 d in vitro and then double treated for 24 h with GA and DL (each 10 mM). Recruitment-withdrawal of FUS-eGFP to UV laser cuts in nuclei (boxed area) was then imaged live (Video 4) similar to (C). Shown are single video frames at 150 s when the eGFP intensity was around its maximum. Note the failed recruitment in FUS Mock as compared with Ctrl Mock and its rescue through GA and DL double treatment. Scale bar = 10 µm. **(F)** Quantification of (E), amount of FUS-eGFP at cut over time. Note the failed recruitment in FUS Mock (red curve) over the entire recording time of 850 s whereas GA and DL double treatment (green curve) fairly restored the recruitment-withdrawal towards Ctrl kinetics (blue curve).

PARP1 (Mastrocola et al, 2013; Rulten et al, 2014), which is impaired in the case of ALS-causing mutations in the *FUS* gene (see Figs 2 and S4 and [Naumann et al, 2018]). Poly(ADP-ribose) polymerases belong to the three main enzymes consuming NAD⁺. Mammalian cells have evolved a NAD⁺ salvage pathway capable to resynthesize NAD⁺. The products of the glyoxylase DJ-1—GA and DL—can

be reintroduced into metabolic pathways. D-lactic acid can be converted to pyruvate reducing NAD(P)⁺ to NAD(P)H via D-Lactate dehydrogenase, whereas GA converted to glyoxylate (for further use in the citrate cycle) also reduces NAD(P)⁺ to NAD(P)H. Therefore, we hypothesized that supplementation with GA and DL restores NAD⁺ using this salvage pathway and therefore the NAD⁺

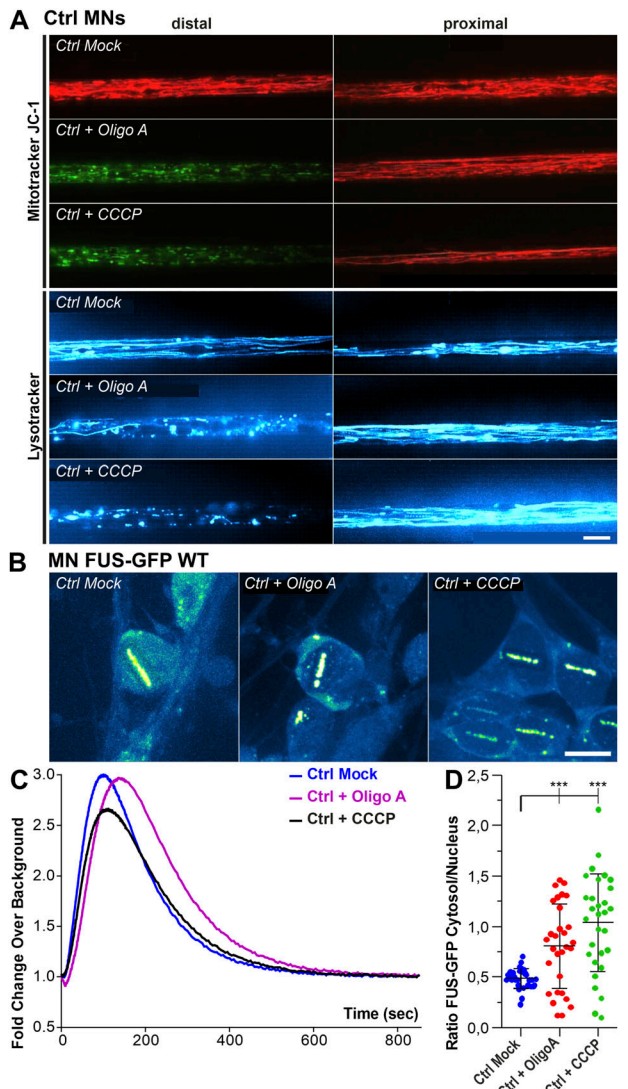

**Figure 3. Site-specific inhibition of mitochondrial ATP production causes local disruption of axonal organelle trafficking.**
**(A)** Patient-derived spinal MNs from healthy control patients were matured for 21 d in vitro in microfluidic chambers. Then 10 µM of Oligo A or CCCP was exclusively added to the distal site 4 h before live imaging at the distal (left) versus proximal (right) channel end with Mitotracker JC-1 (red/green) (Video 5) and Lysotracker (cyan hot) (Video 6). Shown are single, representative microchannels of the microfluidic chamber microgroove barrier either at the distal (left) or proximal (right) end, which were inhabited by a protruding bundle of typically 5–20 axons. Axonal organelle motility is visualized by maximum intensity projections of videos. Processive motility results in straight, longer trajectories whereas immobile organelles project as punctae. Note the drastic loss of lysosomal and mitochondrial motility and its inner membrane potential (JC-1 green) at the treated distal site (Oligo A, CCCP) compared with Mock whereas the untreated proximal site remained physiological. Shown is a representative example of Ctrl1 (Table 1). Scale bar = 10 µm. **(B)** Patient-derived spinal MNs expressing normal (Ctrl) FUS-eGFP WT (Table 1) were matured for 21 d in vitro in uncompartmentalized dishes. Cells were treated for 4 h with 10 µM Oligo A or CCCP, recruitment-withdrawal of FUS-eGFP to UV laser cuts in nuclei was then imaged live (Video 7). The eGFP intensity is shown in the LUT "Green Fire Blue" of the FIJI software, i.e., low eGFP intensities are shown in blue and high intensities in green shades, no nuclear HOECHST staining or alike was used. Each viewing field centres single nuclei with their surrounding cytosol. **(C)** Quantification of (B), amount of FUS-eGFP at cuts plotted over time. Note that neither Oligo A (purple curve) nor CCCP (black curve) inhibited the normal FUS recruitment (blue

precursor nicotinamide riboside (NAR) (Harlan et al, 2016, 2020) leads to a similar rescue effect as GA and DL.

As a test, we supplemented the bacterial artificial chromosome HeLa FUS model (Table 1) either with NAR or inhibited the rate limiting enzyme of NAD(P)$^+$ synthesis, nicotinamide phosphoribosyltransferase (NAMPT), using FK866. NAR supplementation rescued FUS cytoplasmic mislocalization (Fig 4A and B) and also FUS recruitment to DNA damage sites in the FUS-eGFP P525L mutant (Fig 4C) in a dose-dependent manner (Fig 4E). Furthermore, NAR increased the abundance of FUS at DNA damage sites in WT cells (Fig 4D). Conversely, inhibition of NAMPT with FK866 led to a FUS-ALS phenocopy with cytoplasmic mislocalization of FUS (Fig 4A and B) and dose-dependent loss of recruitment of FUS to sites of DNA damages in WT cells (Fig 4C and F), although it had no additional effect on mutant FUS cells (Fig 4C and G).

## GA and DL restore axonal organelle phenotypes in SOD1- but not in TDP43-ALS mutants

We finally wished to test whether the potential therapeutic effects of GA & DL are also seen in other monogenetic ALS mutants. To this end, we chose SOD1- and TDP43-mutant iPSC-derived spinal MNs as these have been reported to show axonal organelle phenotypes at DIV 21 as well, albeit clearly distinct ones (Kreiter et al, 2018; Pal et al, 2018; Gunther et al, 2022): whereas mutant SOD1 MNs exhibited a hyper-elongation of mitochondria along with a reduction in the inner membrane potential but no alteration of mitochondrial and lysosomal speed and track displacement (Gunther et al, 2022), mutant TDP43 MNs displayed a striking decrease in speed and track displacement of both types of organelles in distal and proximal axons but no perturbed mitochondrial elongation and inner membrane potential (Kreiter et al, 2018; Pal et al, 2018). Treatment with 1 mM GA and DL for 24 h was able to restore the axonal mitochondrial phenotypes of SOD1 mutant patient-derived spinal MNs in both the distal and proximal axon parts (Figs 5A, D, and E and S1) but not those of mutant TDP43 spinal MNs (i.e., reduced distal and proximal organelle track displacement and mean speed, Figs 5A–C and S1). Specifically, adding 1 mM GA and DL restored the mitochondrial hyper-elongation and depolarization in SOD1 mutant MNs (Fig 5A, D, and E). Thus, responsiveness to GA and DL seems to be associated with mitochondrial depolarization, as shown for our FUS (Figs 1A and E and S1C) and SOD1 (Fig 5A and E and S1C) mutants.

## Riluzole did not restore mitochondrial phenotypes in FUS- and SOD1- ALS mutant MNs

We finally investigated whether the "gold standard" treatment of ALS, Riluzole, has similar beneficial effects on axonal organelle

---

curve). **(D)** Quantification of (B) of the ratio of FUS-eGFP cytosolic over nuclear integral intensity, N = 30 images, each data point of the scatter dot plots presents one mean value per image, whiskers show the STDEV, centre lines the median. A one-way ANOVA with Bonferroni post hoc test was used for the normal distributions of the data sets to reveal significant differences in pairwise comparisons as highlighted by brackets above data. Asterisks: highly significant alteration in indicated pairwise comparison, ***$P \leq 0.001$.

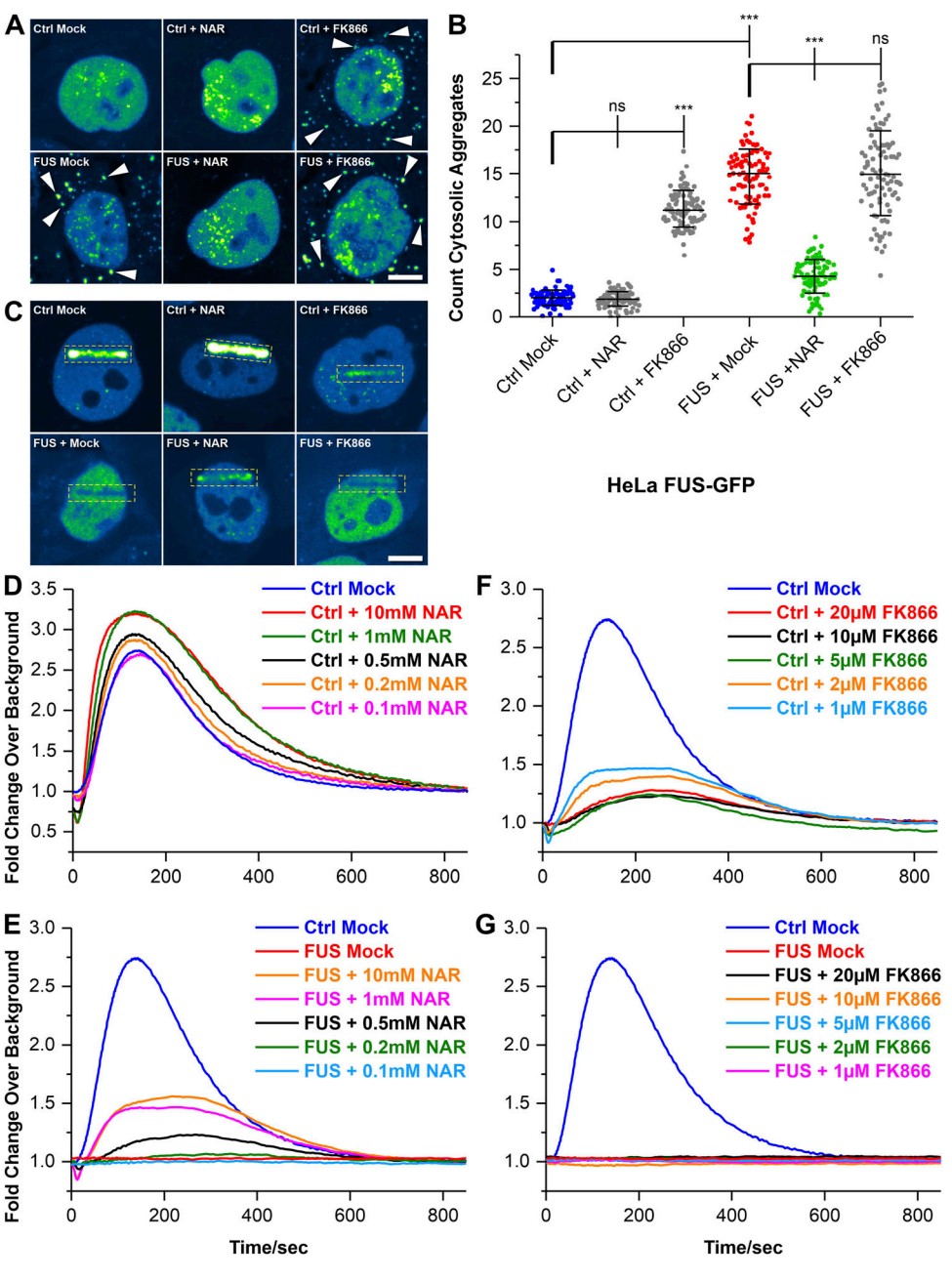

**Figure 4. Recruitment of FUS to nuclear DNA damage sites depends on proper NAD metabolism.**

**(A)** Shown are maximum intensity projections of confocal Z-stacks imaged live in the transgenic bacterial artificial chromosome HeLa cell model expressing normal FUS-eGFP WT (Ctrl) or mutant FUS-eGFP P525L (FUS) (Table 1). The eGFP intensity is shown in the LUT "Green Fire Blue" of the FIJI software, i.e., low eGFP intensities are shown in blue and high intensities in green shades, no nuclear HOECHST staining or alike was used. Each viewing field centres a single nucleus with its surrounding cytosol. The cytosol particularly in Ctrl cells appears very dark similar to intercellular blank background because of its low content of eFUS-GFP. Note the occurrence of cytosolic FUS aggregates in FUS Mock as compared with Ctrl Mock (arrowheads) that were rescued through treatment with 10 mM nicotinamide riboside (NAR) over 24 h, a precursor of NAD⁺. Conversely, inhibition of NAD synthesis through treatment with 10 μM FK688 over 24 h phenocopied the cytosolic mutant FUS aggregation. Scale bar = 10 μm.
**(B)** Quantification of (A) as counts of cytosolic FUS aggregates per cell, N = 90 images, each data point of the scatter dot plots presents one mean value per image, whiskers show the STDEV, centre lines the median. Note the drastic increase in FUS Mock (red dots) as compared with Ctrl Mock (blue dots) and its reversion back to Ctrl levels through NAR treatment (greendots) whereas FK866 treatment of Ctrl cells led to increased cytosolic FUS aggregation similar to mutant FUS Mock. A one-way ANOVA with Bonferroni post hoc test was used for the normal distributions of the data sets to reveal significant differences in pairwise comparisons as highlighted by brackets above data. Asterisks: highly significant alteration in indicated pairwise comparison, ***$P \leq 0.001$ and ns: no significant difference.
**(C)** Transgenic bacterial artificial chromosome HeLa cells from (A) were treated for 24 h with either NAR or FK866. Recruitment-withdrawal of FUS-GFP to UV laser cuts in nuclei (boxed area) was then imaged live (Video 8). Shown are single video frames at 150 s when the GFP intensity was around its maximum. The eGFP intensity is shown in the LUT "Green Fire Blue" of the FIJI software, i.e., low eGFP

intensities are shown in blue and high intensities in green shades, no nuclear HOECHST staining or alike was used. Each viewing field centres a single nucleus with its surrounding cytosol. The cytosol particularly in Ctrl cells appears very dark similar to intercellular blank background because of its low content of FUS-eGFP. Note the failed recruitment in FUS Mock as compared with Ctrl Mock and its rescue through NAR treatment whereas FK866 treatment of Ctrl cells decreased the FUS recruitment. Furthermore, in case of failed FUS-eGFP recruitment (i.e., FUS Mock, FUS + FK866), the laser beam left a dark line because of photo bleaching that is not to be mistaken for FUS-eGFP withdrawal from the DNA damage site. Scale bar = 10 μm. **(D)** Quantification of (C) for Ctrl cells, GFP intensities at cuts plotted over time after laser irradiation. Note the further boosted recruitment of FUS over Ctrl Mock (blue curve) through NAR treatment in a concentration-dependent manner. **(E)** Quantification of (C) for mutant FUS cells. Note the rescue of FUS recruitment over Mock (red curve) through NAR treatment in a concentration-dependent manner, even though not fully restored to Ctrl Mock levels (blue curve). **(F)** Quantification of (C) for Ctrl cells. Note the pronounced decrease in FUS recruitment through FK866 treatment in a concentration-dependent manner, even though no complete inhibition as in FUS Mock was achieved. **(G)** Quantification of (C) for mutant FUS cells. Note that FK866 treatment did not alter the failed FUS recruitment at any concentration.

phenotypes in FUS- and SOD1-patient-derived MNs. To this end, we treated FUS- and SOD1- mutant iPSC-derived spinal MNs with 10 μM Riluzole over the entire maturation period in MFCs until 21 DIV prior imaging. Interestingly, Riluzole had no effect on any of the parameters investigated, which included mitochondrial and lyso-somal trafficking (Fig S7A–C), mitochondrial inner membrane potential (Fig S7E) and mitochondrial fragmentation (FUS-ALS) or elongation (SOD1-ALS) (Fig S7D).

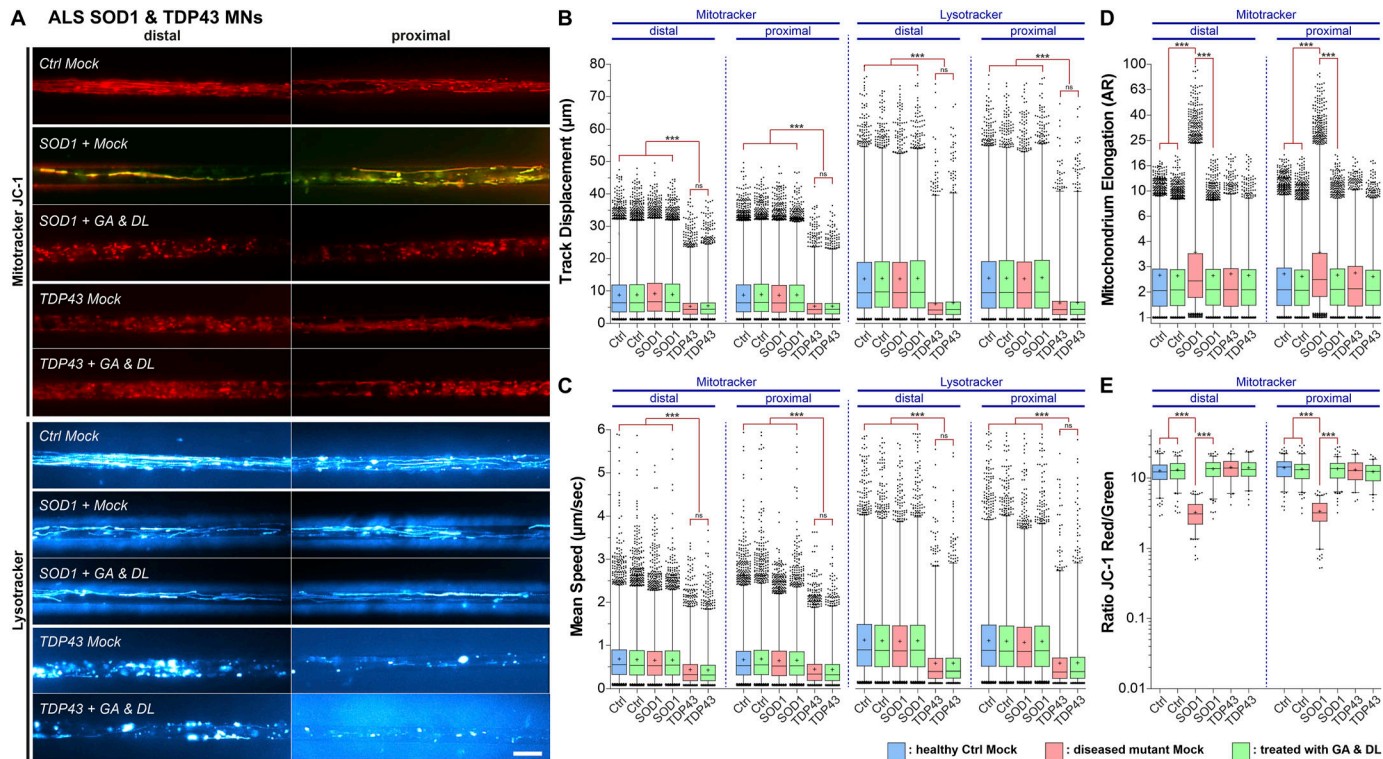

**Figure 5.  GA and DL together rescue mitochondrial hyper-elongation and inner membrane potential in amyotrophic lateral sclerosis-SOD1 axons but not the mitochondrial and lysosomal trafficking defects in TDP-43 mutants.**

Patient-derived spinal MNs were matured for 21 d in vitro in microfluidic chambers, double-treated for 24 h at both sites (distal and proximal) with GA and DL (each 1 mM) and imaged live at the distal (left) versus proximal (right) channel end with Mitotracker JC-1 (red/green) and Lysotracker (cyan hot). **(A)** Maximum intensity projections of videos visualize organelle moving tracks in axons. Shown are single, representative microchannels of the microfluidic chamber microgroove barrier either at the distal (left) or proximal (right) end, which were inhabited by a protruding bundle of typically 5–20 axons. Processive motility results in straight, longer trajectories whereas immobile organelles project as punctae. For mitochondria in SOD1, the first frame of the video is shown instead of the maximum intensity projection to better document the elongation of these organelles. Representative examples from the mutant SOD1 and TDP-43 (Fig S1, Table 1) and control (Ctrl) line pools are shown as follows: SOD1: SOD1 D90A, TDP-43: TDP-43 S393L, Ctrl: Ctrl1. Note the striking elongation of mitochondria along with a reduction in the inner membrane potential (JC-1 yellow overlap) in SOD1 Mock compared to Ctrl (JC-1 red) at both the distal and proximal site, which were rescued through GA and DL double treatment. Conversely, mitochondrial inner membrane potential appeared normal in TDP-43 Mock (JC-1 red), whereas mitochondrial and lysosomal motility was reduced at both the distal and proximal site (punctae instead of trajectories) and could not be restored through GA and DL double treatment (Video 9 and Video 10). Scale bar = 10 μm. **(B, C, D, E)** Box plots quantifications of various tracking and morphology parameters deduced from videos from (A) as per organelle values (i.e., each data point presents one individual organelle) for the mutant SOD1, TDP-43, and Ctrl cell line pool, except of (E) showing mean values per image. For mutant SOD1, data from the SOD1 D90A, A4V, and R115G lines were pooled (Table 1). For mutant TDP-43, data from the TDP-43 S393L and G294V lines were pooled (Table 1). For WT Ctrl, data from the Ctrl1, Ctrl2, Ctrl3, FUS WT-GFP, and SOD1 D90A igc lines were pooled (Table 1). For individual cell lines refer to Fig S1. Box: 25–75% interquartile range, horizontal line: median, cross: mean, whiskers: non-outlier range (99% of data), dots outside whiskers: outliers, Ctrl Mock is shown in pale blue, diseased mutant mock in pale red, double treatment with GA and DL in pale green. **(B, C, D, E)** A one-way ANOVA with either a Kruskal-Wallis post hoc test to account for the non-normal, top-tailed data distributions (B, C, D) or Bonferroni post hoc test for the normal distributions (E) was used to reveal significant differences in pairwise comparisons. Asterisks: highly significant alteration in pairwise comparisons as highlighted by brown brackets above data, ***P ≤ 0.001, all other pairwise comparisons were not significantly different. **(B, C)** Note the drastic reduction in distal as well as proximal track displacement (B) and mean speed (C) in TDP-43 Mock that was not rescued through GA and DL double treatment. **(D)** Note the drastic elongation of mitochondria at both the distal and proximal site in SOD1 that was rescued by GA and DL double treatment. **(E)** Note the reduction in mitochondrial inner membrane potential in SOD1 at both the distal and proximal site that was rescued by GA and DL double treatment.

## Discussion

In this study, we introduce the two substances glycolic acid (GA) and D-lactate (DL) as novel therapeutic candidates for ALS. Both compounds occur naturally in the cell, e.g., as products of DJ-1 (Lee et al, 2012), which converts the reactive aldehydes glyoxal and methylglyoxal to GA and DL, respectively (Thornalley, 2003; Lee et al, 2012). We show that combinatorial treatment with GA and DL restored axonal organelle deficits of mitochondria and lysosomes in FUS- and SOD1-ALS. Specifically, in the case of FUS mutants, GA and

DL restored the axonal motility of distal lysosomes and mitochondria (Fig 1B and C) as well as the fragmentation and depolarization of distal mitochondria (Fig 1D and E). Conversely, the SOD1 mutants did not exhibit any alteration in their axonal organelle motility (Fig 5B and C) but instead a striking hyper-elongation of their axonal mitochondria along with a reduced inner membrane potential (Fig 5D and E), albeit to a lesser extent as compared with the FUS mutants (Fig S1C). These distinct axonal organelle phenotypes in mutant SOD1 were restored through GA and DL treatment as well. Because the common denominator of the mutant FUS

and SOD1 phenotypes was apparently the reduced inner membrane potential of mitochondria, we suggest that the mode of action of GA and DL in rescuing these distinct phenotypes functions through the restoration of mitochondrial polarization. In line with this view is our finding that GA and DL failed to restore the deficient axonal organelle motility in TDP43 mutants that did not exhibit any alteration in their mitochondrial inner membrane potential (Fig 5E). This fits previous data on *PARK7* cells and *C. elegans* models, in which GA restored the mitochondrial membrane potential and prolonged neuronal survival (Toyoda et al, 2014). It remains, however, open, why in case of ALS mutants, only a combinatorial treatment of GA and DL was able to restore the phenotypes in our iPSC-derived spinal motoneuronal ALS model.

GA and DL can help to maintain calcium homeostasis (Chovsepian et al, 2022). ER-mitochondria associations have become of increasing interest in neurodegenerative diseases because these specialized tight structural associations between a closely apposed ER surface and outer mitochondria membrane were reported to regulate a variety of essential physiological functions including calcium signaling, phospholipid synthesis/exchange, mitochondrial biogenesis, and dynamics as well as cell death (Grossmann et al, 2019; Liu et al, 2019; Grossmann et al, 2023; Pereira et al, 2023). Interestingly, loss of DJ-1 led to reduced ER-mitochondria association and disturbed function of mitochondria-associated membranes and mitochondria in vitro (Liu et al, 2019). Whether the beneficial effects of GA and DL in FUS- and SOD1-ALS are attributable to improved mitochondria-ER interactions requires, however, further investigations.

Apart from rescuing mitochondrial depolarization and phenotypes, GA and DL did also restore FUS recruitment to laser induced DNA damage sites (Fig 2). We recently showed that the lack of proper FUS-recruitment to DNA damage sites is upstream of all axonal/mitochondrial phenotypes in FUS-ALS (Naumann et al, 2018). DNA damage induces PARP1, an enzyme important in initiating proper DNA damage response. Activation of PARP1, however, not only leads to NAD$^+$ depletion but can also induce mitochondrial dysfunction (Szabo et al, 1996). Silencing of PARP1 increased basal cellular parameters of OXPHOS, providing direct evidence that PARP1 is a regulator of mitochondrial function in resting cells. Whereas PARP1 is a regulator of OXPHOS in resting and oxidatively stressed cells, it only exerts a minor effect on glycolysis (Modis et al, 2012). Interestingly though, energy depletion itself is not sufficient to induce DNA damage (Szabo et al, 1996). This perfectly fits our data showing that administration of Oligomycin A or CCCP led to mitochondrial depolarization and halted axonal transport, whereas it did not influence laser-induced FUS-recruitment (Fig 3B and C). These data might suggest that the lack of proper DNA repair in FUS-ALS leads to a sustained PARP1 activation and NAD$^+$ degradation.

Glyoxylases are important to detoxify, e.g., methylglyoxal (MGO) and glyoxal (GO), which are generated, e.g., during glycolysis. If not detoxified, advanced glycation end products (AGEs) are accumulating. Inceased levels of AGEs are associated with aging and with diverse disorders such as diabetes, renal failure and neurodegeneration. Lipid peroxidation and AGEs occur in the brain during normal aging as well as in Alzheimer's disease (Dei et al, 2002) and also in ALS patient spinal cord motor neurons (Kikuchi et al, 2000, 2002). FUS-ALS mutations led to increased ROS production and lipid

peroxidation as well (Ismail et al, 2024). AGE Nε-(carboxymethyl) lysine levels are elevated in cerebrospinal fluid of ALS patients (Kaufmann et al, 2004). This suggests that impaired glyoxylase activity might be an important contributor to ALS pathophysiology as well. The generation of MGO and GO is further augmented by ROS. The transcription factor NRF2 is a critical inducer of the antioxidant response element (ARE)-mediated gene expression, and, importantly, is regulated by DJ-1. Overexpression of DJ-1 results in increased NRF2 protein levels by preventing association with its inhibitor protein, KEAP1, and subsequent ubiquitination of NRF2 (Clements et al, 2006). It is of note that increased NRF2 levels protect against DNA damage by activating DNA damage repair factors. Interestingly, previous data of SOD1 mice showed that the toxicity of astrocytes expressing ALS-linked mutant hSOD1 to co-cultured motor neurons was reversed by NRF2 overexpression (Vargas et al, 2008). However, GA and DL most likely act downstream of DJ-1 and NRF2 or independent of the latter.

We thus wondered whether effects of GA and DL were because of alleviation of impaired metabolism by reintroducing them into metabolic pathways. DL can be converted to pyruvate, reducing NAD(P)$^+$ to NAD(P)H, whereas GA converted to glyoxylate (for further use in the citrate cycle) also reduces NAD(P)$^+$ to NAD(P)H. NAD$^+$ is an important metabolite in human cells pivotal for processes including DNA repair and mitophagy (Hou et al, 2021). A lot of metabolic and stress pathways oxidize NAD(P)H including glycolysis, but also ROS scavenging by the gluthation system, and PARP1 activation. Furthermore, enzymes involved in NAD$^+$ salvage, namely NAMPT and NMNAT2, were reported to show an altered expression in the spinal cord of ALS patients, suggesting deficits of this pathway in the human ALS pathology (Harlan et al, 2020). Therefore, we hypothesized that supplementation with GA and DL restores NAD$^+$ using this salvage pathway and, thus, boosting this salvage pathway with the NAD$^+$ precursor NAR should have similar effects as GA and DL treatment (Harlan et al, 2016, 2020). Fitting to this theory, inhibition of NAMPT perfectly phenocopied FUS-ALS mutants, whereas supplementation with NAR rescued FUS-ALS phenotypes (Fig 4). Of note, enhancing the NAD$^+$ salvage pathway was recently reported to revert the toxicity of astrocytes expressing ALS-linked mutant hSOD1 to co-cultured MNs (Harlan et al, 2016). Conversely, knock out of DJ-1 in the G93A SOD1 ALS mouse model led to an accelerated disease course and shortened survival (Lev et al, 2015). In addition, it has been recently shown that increased demand for NAD$^+$ relative to ATP induces aerobic glycolysis (Luengo et al, 2021). We recently showed that a boosted metabolic turnover of the glycolytic pathway improved the viability of FUS-ALS MN, whereas blocking glycolysis reduced their viability (Zimyanin et al, 2023). These data altogether suggest that the beneficial treatment effect of GA and DL in FUS- and SOD1-ALS MNs might be because of a metabolic rescue by restoring the NAD(P)H reservoir.

A limitation of the results is that rather high concentrations of the compounds GA and DL were required for the rescue effects. The natural compound D-lactate is found in the body at concentrations of about 10–20 μM in the blood serum (Hasegawa et al, 2003). Physiological serum concentrations of glycolate are up to 12.5 μM and a bit higher in tissues (also in brain tissue)

(Hagen et al, 1993; Knight et al, 2012). The concentrations found, e.g., in diabetic patients are even higher; however, GA and DL are thought to represent the products of detoxified MGO and GO, respectively, and not the toxic agents themselves. Thus, the respective EC50 found in our study were considerably higher and thus future evaluation of their safety are needed. Another limitation is that the MFCs contained axon bundles rather than single axons. We thus cannot rule out that single axons behave different. Axonal growth behavior was also not assessed by these assays, but analysis was performed in very standardized proximal and distal positions.

In addition, we did not yet understand why for the rescue effect in the case of FUS- and SOD1-ALS both substances were needed, whereas only one was sufficient in case of PD (Toyoda et al, 2014) or ischemia (Chovsepian et al, 2022). We hypothesize that GA's role is to provide substrate for the citrate cycle whereas DL provides the energy. Even with a lot of energy, this cannot be converted to ATP if the citrate cycle does not have sufficient capacity. However, whether this is different in PD versus ALS models needs to be clarified in further studies. Furthermore, the role of DJ-1 in SOD1- and particularly FUS-ALS, and whether the effects of GA and DL might be a sign of impaired DJ-1 function in SOD1- and/or FUS-ALS needs to be addressed in future studies. Finally, under certain conditions, DJ-1 can produce a mixture of D-lactate and L-lactate (Zhou et al, 2022). However, L-lactate was neither effective alone nor in combination with GA in the FUS-ALS MNs. The reason for this might be the different cellular distribution of D- versus L-lactate dehydrogenases. The L-enantiomer is converted by LDH-A or LDH-B, which are both located in the cytoplasm. The resulting pyruvate is then required to be transported into the mitochondrion. In contrast, the D-enantiomer is converted by LDHD—which is a mitochondrial protein. Thus, DL must be directly imported into the mitochondrion (de Bari et al, 2002; Flick & Konieczny, 2002). D- and L-lactate were shown to enter mitochondria, but in a stereospecific manner (de Bari et al, 2002). A recent study, however, reported that DJ-1 converts MGO to D- and L-lactate in two different manners. Either DJ-1 acts as a glyoxalase and stereoselectively transfers MGO to D-lactate, which requires glutathione (GSH) as cofactor. Alternatively, DJ-1 acts as a protein deglycase, by which it is able to rescue MGO-modified peptides/proteins in the absence of GSH, resulting in the production of both D- and L-lactate (Zhou et al, 2022). Because there is no lack of GSH in case of FUS-ALS mutants (Ismail et al, 2024), we speculate that only the former is of relevance in our cell model. However, this hypothesis awaits its verification in future studies.

In summary, we present novel insights into the pathophysiology of SOD1- and particularly FUS-ALS, revealing a putatively central role for glycolic acid and D-lactate. We also show that, albeit presenting an early axonal transport deficiency as well, TDP43 patient-derived MNs did not share this mechanism. This points towards the necessity of individualized (gene-) specific therapy stratification. Our findings also suggest that mitochondrial depolarization (found, e.g., in FUS and SOD1-ALS, but also DJ-1-PD and others) might be a common drug target. GA and DL might thus constitute interesting novel drug candidates in subsets of ALS cases and feasibly other neurodegenerative diseases suffering from mitochondrial depolarization.

# Materials and Methods

### Characteristics of patients for iPSC derivation

We studied iPSC-derived spinal MN cell cultures from familiar ALS patients with the following pathogenic mutations (Mt): TDP43 S393L[het], TDP43 G294V[het], SOD1 D90A[hom], SOD1 A4V[het], SOD1 R115G[het], FUS R521C[het], FUS R521L[het], FUS R495QfsX527[het] and compared them with MNs carrying human WT counterpart alleles in three cell lines from healthy volunteers (WT, Ctrl1-3), and a gene-corrected isogenic control (IGC) line of SOD1 D90A (SOD1 D90A igc). Moreover, parental FUS R521C was used to generate isogenic FUS-P525L eGFP and its gene-corrected control FUS-WT eGFP (Naumann et al, 2018). All cell lines were obtained by skin biopsies of patients and healthy volunteers and have been described before (Japtok et al, 2015; Naujock et al, 2016; Kreiter et al, 2018; Naumann et al, 2018; Bursch et al, 2019) (Table 1). The performed procedures were in accordance with the Declaration of Helsinki (WMA, 1964) and approved by the Ethical Committee of the Technische Universität Dresden, Germany (EK 393122012 and EK 45022009). Written informed consent was obtained from all participants for publication of any research results.

### Genotyping

DNA from the cell lines was genotyped by a diagnostic human genetic laboratory (CEGAT). Control lines were also genotyped and did not show any ALS-associated mutation.

### Mycoplasma testing

Every cell line was checked for mycoplasma when entering the lab and after reprogramming. Routine checks for mycoplasma were done every 3–6 mo. We used the Mycoplasma Detection kit according to manufacturer's instructions (No 11–1025; Firma Venor GeM).

### Generation, gene-editing, and differentiation of human iPSC cell lines to MNs in MFCs

The generation and expansion of iPSC lines from healthy control and familiar ALS patients with defined mutations in distinct ALS genes (Table 1) were recently described (Japtok et al, 2015; Naujock et al, 2016; Naumann et al, 2018). The gene-corrected isogenic control to the homozygous mutant SOD1 D90A (SOD1 D90 igc, Table 1) was generated by CRISPR/-Cas9-mediated gene-editing and fully characterized (Bursch et al, 2019). To generate the two isogenic cell lines FUS-WT eGFP and FUS-P525L eGFP, the FUS R521C line was used as parental source (Table 1). The patient-specific FUS R521C mutation was altered at its mutation site and simultaneously C-terminally tagged with eGFP by CRISPR/Cas9-mediated genome editing to obtain a P525L mutation instead along with a gene-corrected WT control. Both new lines were fully characterized (Naumann et al, 2018). The subsequent differentiation of all iPSC lines to neuronal progenitor cells (NPC) and further maturation to spinal MNs was described previously (Reinhardt et al, 2013; Naumann et al, 2018). Specifically, the differentiation pipeline from

NPCs to mature MNs is illustrated in Fig 1A of Naumann et al (2018), in Fig 1A of Kreiter et al (2018) and in Fig 4A of Gunther et al (2022). The coating and assembly of MFCs (Xona Microfluidics RD900) to prepare for the seeding of MNs was performed as described (Naumann et al, 2018; Pal et al, 2018, 2021). MNs were seeded for maturation into one site of a MFC to obtain a fully compartmentalized culture with proximal somata and their dendrites being physically separated from their distal axons as only the latter type of neurite was capable to grow from the proximal seeding site through a microgroove barrier of 900 µm-long microchannels to the distal site where they sprouted out from the distal channel exits to the open space (shown in details in Glaß et al [2020]). Specifically, the schematic set-up of compartmentalized MFC cultures is illustrated in Fig 2A of Naumann et al (2018), in Fig 1, 2A, and 4A of Pal et al (2018), in Fig 1A of Pal et al (2021) and in Fig 4B of Gunther et al (2022). All subsequent imaging in MFCs was performed at DIV 21 of axon growth and MN maturation (DIV 0 = day of seeding into MFCs). All images and corresponding videos in this report of MNs in MFCs show representative sections of single microchannels either from their distal or proximal end within the microgroove barrier at direct juxtaposition to the channel proximal entry or distal channel exit, respectively (Figs 1A, 3A, and 5A, S2A, and S7A). These images and corresponding videos depict whole bundles typically comprising 5–20 axons. It was technically not possible to discern between individual axons within these bundles or to trace a single axon from its soma in the proximal seeding chamber through the microchannel to its growth cone in the distal open chamber because of the optical resolution limits of the microscope and the overall spatial, dense complexity of the whole compartmentalized architecture of the MFC culture. Therefore, it was not possible to correlate the deduced organelle tracking and morphology parameters (see below) with the total axon length or alike. However, this limitation was of no relevance for the bulk organelle tracking analysis and the biological questions addressed in this study. Finally, we verified in our previous reports (Naumann et al, 2018; Glaß et al, 2020) that the neurites within the microchannels and beyond the distal exits where completely pure for motoneuronal axons with no connecting downstream cells such as myotubes at their distal growth cones, i.e., we did not establish any composite cultures.

**Live imaging of MNs in MFCs**

Time-lapse video acquisition was performed as described previously (Naumann et al, 2018; Pal et al, 2018). In brief, to track lysosomes and mitochondria, cells were double-stained with live cell dyes Lysotracker Red DND-99 (Cat. No. L-7528; Molecular Probes) and Mitotracker Deep Red FM (Cat. No. M22426; Molecular Probes) at final concentrations of 50 nM each. Trackers were added from a 1 mM stock in DMSO directly to culture supernatants and incubated for 1 h at 37°C. Live imaging was then performed without further washing of cells in the Center for Molecular and Cellular Bioengineering, Technische Universität Dresden (CMCB) light microscopy facility on a Leica HC PL APO 100x 1.46 oil immersion objective on an inversed fluorescent Leica DMI6000 microscope enclosed in an incubator chamber (37°C, 5% $CO_2$, humidified air) and fitted with a 12-bit Andor iXON 897 EMCCD camera (512 × 512 pixel, 16 µm/pixels on chip, 229.55 nm/pixel at 100x magnification

with intermediate 0.7X demagnification in the optical path through the C-mount adapter connecting the camera with the microscope). For more details, refer to https://www.biodip.de/wiki/Bioz06_-_Leica_AFLX6000_TIRF and our previous publications (Naumann et al, 2018; Pal et al, 2018). Fast dual color videos were recorded at 3.3 frames per second (fps) per channel over 2 min (400 frames per channel in total) with 115 ms exposure time as follows: Lysotracker Red (excitation: 561 nm laser line, emission filter TRITC 605/65 nm) and Mitotracker Deep Red (excitation: 633 nm laser line, emission filter Cy5 720/60 nm). Dual channel imaging was achieved sequentially by fast switching between both laser lines and emission filters using a motorized filter wheel to eliminate any crosstalk between both trackers.

**Organelle tracking and shape analysis of live imaging videos**

Recently, we have published a comprehensive description of the automated analytical pipeline starting from object recognition in raw video data to final parametrization of organelle motility and morphology (Pal et al, 2018). In brief, organelle recognition and tracking was performed with the FIJI Track Mate plugin, organelle shape analysis with our custom-tailored FIJI Morphology macro that is based on the FIJI particle analyzer. Both Track Mate and particle analyzer tools returned the mean speed and track displacement for each organelle type (Mito- versus Lysotracker-labeled) along with the elongation of mitochondria expressed as its aspect ratio (AR), i.e., the ratio of the major over the minor radius of the fitted ellipse. Subsequent data mining of individual per-video result files was performed in KNIME to assemble final results files with annotated per-organelle parameters, thereby allowing all data from each experimental condition to be pooled (e.g., all data for mitotracker or lysotracker at the distal channel readout position for a given cell line under a specific treatment condition such as GA and DL). Data per organelle were visualized as box instead of scatter dot plots because the underlying data sets often comprised ten thousands of pooled organelles (Figs 1B–D and 5B–D, S1A, B, and D, S2B–D, S3A, and S7B–D), i.e., each data point presents one organelle except of Figs 1E and 5E, S1C, and S7E (ratio red/green Mitotracker JC-1, see below) that show averaged per-image data. Data for all box plots were pooled from four independent experiments.

**Analysis of inner mitochondrial membrane potential (ratio JC-1 red/green channel)**

Analysis of mitochondrial membrane potential with Mitotracker JC-1 (Cat. No. M34152; Molecular Probes) was performed as described previously (Naumann et al, 2018). In brief, object segmentation was performed with the channel of higher intensity (most often red emission) to generate a selection limited to mitochondria using a custom-tailored FIJI macro. The resulting selection was saved as a region of interest (ROI) and applied to both channels to reveal the total integral intensity and area of mitochondria and background in both channels using the "Measure" command. After area normalization and background subtraction, ratios of integral red/green intensity were taken as mean membrane potential per video (first frame only) and batch-analyzed in KNIME as for the tracking analysis (see above). The resultant ratios were displayed as box

plots of all pooled images on a log scale with each data point presenting one mean value per image (Figs 1E and 5E, S1C, and S7E).

## Image/video quantification

For cytosolic FUS-eGFP mislocalization in HeLa cells (Figs 2B and 4B and S5B) and MNs (Figs 3D and S6B), three independent experiments were performed and at least 10 confocal Z-stacks per experiments analyzed as described previously (Naumann et al, 2018). In brief, standard tools of FIJI software were used to measure fluorescence integral intensity within the nucleus and cytosol and to determine the count of discrete objects. Resultant data (FUS cytosolic aggregate counts and ratios cytosolic/nuclear integral intensity) were plotted as scatters in which each dot presents one mean value per image (Figs 2B, 3D, and 4B, S5B, and S6B). For video analysis of MFCs (organelle tracking and shape, mitochondrial inner membrane potential), at least 10 videos were acquired of each MFC (= one technical replicate) with three MFCs per experiment and four independent experiments (= MN differentiation pipeline) per cell line, typically resulting in large, pooled bulk data sets of ten thousands of analyzed organelles.

## Statistical analyses of box and scatter dot plots

Statistical analyses were performed using GraphPad Prism version 5.01. The data sets of the organelles' mean speed and track displacement (lysosomes and mitochondria) as well as the mitochondrial elongation exhibited a typical asymmetrical, top-tailed distribution consistent with our previous reports (Naumann et al, 2018; Pal et al, 2018) and, thus, did not pass the D'Agostino-Pearson test for normality (i.e., they did not obey a Gaussian distribution). Therefore, to test for significant differences between multiple groups, a one-way ANOVA followed by the Kruskal-Wallis post hoc test for pairwise comparisons was used. Conversely, the data sets of the inner mitochondrial membrane potential (mitotracker JC-1) as well as for the cytosolic mislocalisation of FUS passed the normality test above and, thus, the Bonferroni post hoc test in the one-way ANOVAs was performed instead. Alpha < 0.05 was used as the cut off for significance (*$P < 0.05$, **$P < 0.005$, ***$P < 0.001$, ****$P < 0.0001$).

## DNA damage laser irradiation assay

Isogenic FUS-WT eGFP and FUS-P525L eGFP (Table 1) spinal MNs were differentiated from NPCs as described above, finally split and 300,000 cells seeded into uncompartmentalized 3.5 cm dishes instead of MFCs. Dishes were coated before with poly-L-lysine and laminin as described. All subsequent imaging of DNA damage response to laser irradiation sites was performed at DIV 21 of MN maturation (D0 = day of seeding into final dishes) as described previously (Naumann et al, 2018). In brief, a focused 355 nm UV laser beam was directed through a stereotactic galvanometric mirror box to desired x-y-z-positions in cell samples held on a standard inverted Axio Observer Z1 Zeiss microscope equipped with a motorized stage and a piezo-electric Z-actuator. A Zeiss alpha Plan-Fluar 100 × 1.45 oil immersion objective was used and 24 laser shots in 0.5 $\mu$m-steps were administered over 12 $\mu$m linear cuts located within cell nuclei. The cellular response to this DNA damage comprised a fast recruitment of FUS-eGFP to the laser cut site followed by its slower withdrawal (on-off kinetics) and were recorded live over 15 min by confocal spinning disc imaging of the eGFP tag using a 488 nm laser line and a 12-bit Andor iXON 897 EMCCD camera (512 × 512, 16 $\mu$m pixels, 229.55 nm/pixel at 100X magnification) at initial 1 fps and later 0.2 fps during the slower withdrawal phase. For analysis, integral GFP intensity of image selections limited to cuts were determined in FIJI and plotted as fold change over nuclear background (y-axis) over time (x-axis) to reveal on-off kinetics of FUS (Figs 2D and F, 3B and C, and 4D–G, and S4A).

## Treatments and inhibitors

Glycolic acid (GA, Cat. # 124737; Sigma-Aldrich), D-lactate (DL, Cat. # 71716; Sigma-Aldrich) and L-lactate (LL, Cat. # 71718; Sigma-Aldrich) were each dissolved in pure, sterile water to obtain 1 M stocks, respectively. For GA, 6 M NaOH was added drop-wise to assist the dissolution. All stocks were finally sterile-filtrated. For MNs in MFCs, GA and DL or GA and LL were added together 24 h before imaging to both the distal and proximal site at final 1 mM each (Figs 1A and 5A and S2A), unless otherwise stated in the titration experiments (Fig S3A). For laser irradiation experiments and revealing FUS aggregates, GA and DL or GA and LL were added together 24 h before imaging to uncompartmentalized dishes at final 10 mM each (Figs 2A, C, and E and S5), unless otherwise stated in the titration experiments (Fig S4A). For single compound treatments, GA, DL, or LL were each added alone to final 20 mM either to both sites in MFCs or to uncompartmentalized dishes (Figs 1A and 2A and C, S1E, S3A, and S5).

Carbonylcyanid-3-chlorphenylhydrazon (CCCP, Cat. # C2759; Sigma-Aldrich) and Oligomycin A (Oligo A, Cat. # 75351; Sigma-Aldrich) were dissolved in DMSO to obtain a 10 mM stock, respectively. Final working concentrations were 10 $\mu$M for each inhibitor. Each inhibitor was added to uncompartmentalized MNs (Fig 3B) or exclusively to the distal site of MFCs (Fig 3A) just 4 h before imaging to avoid toxic side effects and micro flow progression to the proximal MFC site.

Riluzole (Cat. # R116; Sigma-Aldrich) was dissolved in DMSO to obtain a 10 mM stock for a final working concentration of 10 $\mu$M. Culture supernatants were continuously supplemented at both the distal and proximal MFC site with riluzole over the entire MN maturation of 21 d in MFCs before imaging (Fig S7A).

FK866 (Daporinad, Cat. # S2799; Selleckchem) was dissolved in DMSO to obtain a 20 mM stock for a final working concentration of up to 20 $\mu$M. NAR chloride (NAR, Cat. # SMB00907; Sigma-Aldrich) was dissolved in sterile culture medium at 100 mM for a final working concentration of up to 10 mM. Either FK866 or NAR was added to uncompartmentalized cells 24 h before imaging (Fig 4A and C).

DMSO was used as Mock control for CCCP, Oligo A at final 0.1%. Sterile water was used as Mock control for GA and DL.

# Supplementary Information

# Acknowledgements

We acknowledge the great cell culture help of Sylvia Kanzler, Anett Böhme, Katja Zoschke. The Light Microscopy Facility (LMF) of CMCB (Center for Molecular and Cellular Bioengineering, Technische Universität Dresden) provided excellent support for all live imaging experiments. We thank Ronny Sczech for having programmed the original FIJI/KNIME analytical HC organelle trafficking pipeline. This work was supported, in part, by the NOMIS foundation to A Hermann. A Hermann is supported by the Hermann und Lilly Schilling-Stiftung für medizinische Forschung im Stifterverband. R Günther was supported by niemALSaufgeben.eV and an ALS-family. S Petri and F Wegner were supported by a grant of the Petermax-Müller-Stiftung and the Initiative Therapieforschung ALS e.V. E Storkebaum is supported by an ERC consolidator grant (ERC-2017-COG 770244), and funding from the Radala Foundation, "Stichting ALS Nederland", AFM-Telethon, ARSLA, the "Prinses Beatrix Spierfonds" (W.OR22-03), the Muscular Dystrophy Association (MDA 946876), and an NWO Open Competition ENW-M grant.

## Author Contributions

A Pal: conceptualization, data curation, formal analysis, validation, investigation, visualization, methodology, and writing—original draft, review, and editing.

D Grossmann: resources, data curation, formal analysis, visualization, and writing—review and editing.

H Glaß: resources, data curation, software, formal analysis, visualization, and writing—review and editing.

V Zimyanin: data curation, formal analysis, and writing—review and editing.

R Günther: resources, data curation, formal analysis, and writing—review and editing.

M Catinozzi: data curation, formal analysis, and writing—review and editing.

TM Boeckers: resources and writing—review and editing.

J Sterneckert: resources and writing—review and editing.

E Storkebaum: resources, data curation, and writing—review and editing.

S Petri: resources, data curation, and writing—review and editing.

F Wegner: resources, data curation, and writing—review and editing.

SW Grill: resources, methodology, and writing—review and editing.

F Pan-Montojo: conceptualization, resources, and writing—review and editing.

A Hermann: conceptualization, resources, supervision, funding acquisition, project administration, and writing—original draft, review, and editing.

## Conflict of Interest Statement

A Hermann has received personal fees and non-financial support from Biogen and Desitin during the conduct of the study outside the submitted work. R Günther has received honoraria from Biogen as an advisory board member and for lectures and as a consultant and advisory board member from Hoffmann-La Roche. He also received travel expenses and research support from Biogen. F Pan-Montojo has a patent on the use of glycolic acid and D-lactate as neuroprotective treatment for neurodegenerative disease. In addition, F Pan-Montojo is the CEO and CSO of Neurevo GmbH, a biotech company developing glycolic acid and D-lactate as neuroprotective treatments for several neurological diseases.

## Institutional review board statement

The performed procedures were in accordance with the Declaration of Helsinki (WMA, 1964) and approved by the Ethical Committee of the Technische Universität Dresden, Germany (EK 393122012 and EK 45022009).

## Informed consent statement

Written informed consent was obtained from all participants including for publication of any research results.

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
