## [Reviewer comments · Life Science Alliance]

Life Science Alliance

Glycolic acid and D-lactate-putative products of DJ-1-restore neurodegeneration in FUS and SOD1-ALS

Arun Pal, Dajana Grossmann, Hannes Glaß, Vitaly Zimyanin, Rene Günther, Marica Catinozzi, Tobias Boeckers, Jared Sternecker, Erik Storkebaum, Susanne Petri, Florian Wegner, Stephan Grill, Francisco Pan-Montojo, and Andreas Hermann
DOI: <https://doi.org/10.26508/lsa.202302535>

Corresponding author(s): Andreas Hermann, University of Rostock

Review Timeline:	Submission Date:	2023-12-15
	Editorial Decision:	2024-01-22
	Revision Received:	2024-04-19
	Editorial Decision:	2024-04-25
	Revision Received:	2024-05-05
	Accepted:	2024-05-07

Transaction Report:

January 22, 2024

Re: Life Science Alliance manuscript #LSA-2023-02535-T

Prof. Andreas Hermann
University of Rostock
Gehlsheimer Str. 20
Rostock 18147
Germany

Dear Dr. Hermann,

Thank you for submitting your manuscript entitled "DJ-1 products glycolate and D-lactate restore neurodegeneration in FUS and SOD1- but not TPD43 amyotrophic lateral sclerosis" to Life Science Alliance. The manuscript was assessed by expert reviewers, whose comments are appended to this letter. We invite you to submit a revised manuscript addressing the Reviewer comments.

Thank you for this interesting contribution to Life Science Alliance. We are looking forward to receiving your revised manuscript.

Sincerely,

B. MANUSCRIPT ORGANIZATION AND FORMATTING:

Reviewer #1 (Comments to the Authors (Required)):

Pal and colleagues report on the use of glycolic acid (GA) and D-lactate (DL) to restore defects in organelle trafficking and DNA damage response in cells harboring FUS and SOD1-associated ALS mutations. They use live cell imaging to record mitochondrial and lysosomal trafficking in proximal and distal segments of axons and laser irradiation to induce DNA damage responses and recruitment of FUS. They report that an ipsc-derived motor neuron (MN) line with FUS-R521C mutation has reduced motility (trafficking, displacement) of mitochondria and lysosomes in distal segments of axons. They also report that an ipsc-derived MN line with a TDP43-S393L mutation has reduced motility (trafficking, displacement) of mitochondria and lysosomes in proximal and distal segments of axons. This phenotype in the FUS mutant is reversed by 1 mM GA and DL cocktail (but not with single treatment of GA or DL). GA and DL did not attenuate the mitochondrial phenotype in the TDP43 mutant. In a HeLa cell line and in ipsc differentiated MNs with FUS-P525L mutations, recruitment of GFP-FUS to sites of DNA damage was improved by GA and DL. SOD1-D90A mutants showed abnormal mitochondrial elongation and membrane potential. GA and DL also restored these mitochondrial parameters.

This work has important strengths. The work was hypothesis driven. The group has a variety of human ipsc lines that can be used. A lot of work was performed with the directed differentiation of ipscs to MNs, the live cell imaging, and then the analysis. The figures are generally clear.

This work has important weaknesses.

1. The manuscript as a whole is unfocused. As written, the mitochondrial theme does not fit nicely with the DNA repair theme. The authors need to establish a better logical flow.
2. The different themes (mitochondrial and DNA repair) are both incompletely pursued, despite having many cell lines available. For example, with the different FUS mutations, was distal axon organelle trafficking deficit a common finding? Was DNA repair recruitment abnormally common with all the FUS mutations (they had 5 mutant lines)? Can it be established that those abnormalities in mitochondrial elongation and mitochondrial membrane potential are common among the SOD1 mutants (they had 3 different mutant lines)? Thus, as presented the work appears inconclusive and spotty.
3. The introduction (and the manuscript title) has content related to DJ-1, giving the impression that the following work might have something to do with DJ-1. The authors did not make a good connection with DJ-1. GA and DL can come from sources other than DJ-1. The DJ-1 content could be minimized.
4. Given the number of outliers in the graphs, did the data pass tests for normality?
5. It is not made clear whether each data point in the graphs represents one axon.
6. The mitochondrial and lysosomal trafficking data should be normalized to total axon length. The mutants might have shorter (or longer) axons than controls.
7. The mitochondrial and lysosomal trafficking data could also be normalized to whether that axon has a target contact.
8. The scale bar size should be checked. Or what is shown is a bundle (cable) of axons rather than an individual axon. An unmyelinated axon should be ~ 1 μ m or less.
9. The use of the GA and DL at a concentration of 1 mM is very high for cell culture work. Generally, 100 μ M is considered high. Was there an effect at lower concentrations. If effects were seen at only high concentration, the metabolic interpretation becomes nebulous.
10. Did 20 mM GA or DL cause any evidence of toxicity (Figure 2)
11. In the work presented, what defined the cells as differentiated MNs? For example, the cells shown in Figure 2E and Figure 3B, do not look much like differentiated human MNs. The authors should prepare a supplemental figure defining their human ipsc-derived MN system.
12. The authors should label in the figure panels and graphs whether the results are from HeLa cells or MNs.
13. In Figure 2C and E, why can't the endogenous non-recruited FUS be seen in the nucleus?

Reviewer #2 (Comments to the Authors (Required)):

The authors describe the effects on several cellular parameters of two molecules that are putatively produced by the enzymatic

activity of DJ-1.

The cellular models used are well constructed and represent a wide spectrum of ALS mutations, this aspect is certainly a strong point of the work itself. The quantity of experiments and the quality of these, well characterize the effects of glycolate and D-lactate on axonal mitochondrial transport, on the polarization of the mitochondrial membrane, on the aggregation state of the FUS, etc.

In my opinion, however, the work describes, without giving a mechanistic motivation (for example why are the two molecules effective only in combination?), the action of two molecules that can be produced also through pathways that do not involve the activity of DJ-1. Indeed, the glyoxylase or deglycase activities of DJ-1 are currently widely discussed, and only a few papers, mainly conducted in in vitro models, have suggested such activities among the numerous pleiotropic roles described for DJ-1. Among other things, the putative glyoxylase activity of DJ-1 produces a mixture of D-lactate and L-lactate, and the authors never used both enantiomers.

I therefore find it inappropriate to mention DJ-1 in the title as a major player in the production of such compounds.

At the very least, the authors should have explored the expression levels of DJ-1 in the cell lines used or modified its expression.

In short, I find a weak connection between the experimental structure of the work and the title, which certainly gives different expectations, for example the study of the role of DJ-1 in the ALS patients-derived cell models used.

Reviewer #3 (Comments to the Authors (Required)):

This paper by Pal and colleagues, studies disease mechanisms in iPSC-derived motor neurons from mutant SOD1 and FUS patients. Glycolic acid and lactic acid are shown to counteract some of the phenotypes seen in vitro, in particular deficient axonal trafficking and DNA damage response. The manuscript is of interest to the field, confirms the neuroprotective effect of GA and DA (as previously shown on dopaminergic neurons) and provides avenues for further research.

My comments:

- Abstract: the last paragraph of the abstract is too optimistic. GA and DA have maybe shown positive effects in vitro, but the way to a drug candidate is long. In addition, the concentrations needed are quite high.
- Introduction: the introduction is very long, could be more concise and focused on the research question.
- introduction: references for these statements are missing "DJ-1 overexpression protects dopaminergic neurons against PD, whereas DJ-1 deficiency leads to profound loss of dopaminergic neurons".
- Statistics: one-way ANOVA tests are used when comparing multiple groups, but the p-value shown on the figures is always the comparison with the control I suppose. For mutant conditions with and without treatment, the p-value for this comparison is most relevant and should be displayed as well. This comment is applicable to multiple figures.
- Results figure 2: the mislocalization/cytoplasmic aggregates of FUS are studied in HeLa cells. It would be of interest to add experiments in iPSC-derived motor neurons as well.
- Figure 4: DJ-1 products GA and DL restore nuclear phenotypes by restoring NAD metabolism. The rescue by NAR does not allow to make the claim that the rescue of GA and DL is by restoring NAD metabolism. It can at most suggest it.
- Figure 5: GA and DL together rescue axonal organelle phenotypes in ALS-SOD1 but not in TDP-43 mutants. Only transport data are shown, so the title should maybe be changed.

Reviewer #4 (Comments to the Authors (Required)):

The authors have focussed on the glyoxylase DJ-1 and the downstream products glycolic acid (GA) and D-lactic acid (DL). The authors have supplemented MNs from patient ALS models specifically FUS, SOD1 and TDP43 with GA and DL and measured mitochondrial and lysosomal axonal trafficking, mitochondrial membrane potential, elongation and protein aggregation. Treatment with GA and DL restored axonal trafficking deficits of mitochondria, restored membrane potential and elongation as well as and DNA damage recruitment. The authors have showed that GA and DL combined restore the defects observed in FUS models, partially in SOD1 models but not in TDP43 models.

Overall, certainly in terms of FUS, this is a well thought out approach with robust data sets presented in a transparent manner. The data is clear and the discussion points are valid.

In my opinion the SOD1 data is less clear and somewhat muddles the message. However, this could be clarified with further discussion. The issue arises around the "effect" on axonal trafficking of GL and DL in the SOD1 models. In the results section

the author's state

"GA and DL restore axonal trafficking in SOD1- but not TDP43-ALS mutant"

In the discussion the author's state

"We show that combinatorial treatment with GA and DL restored axonal trafficking deficits of mitochondria and lysosomes in FUS- and SOD1-ALS"

In my opinion GA and DL have no effect on mitochondrial or lysosomal trafficking in SOD1 models. They have a clear effect on mitochondrial membrane potential and shape (response to JC-1 and elongation), but no effect on track displacement and mean speed, probably because the SOD1 MNs don't seem to have defects in the latter two?

Moreover, SOD1 mitochondria are actually more elongated compared to controls compared to FUS which are less elongated. However, GA and DL correct both? So there needs to be careful re-wording of the manuscript and clarity on the precise modes of mechanisms.

One major aspect of the paper overall is that GA and DL alone have no effect at all, which is very surprising. The authors state

"It remains, however, open, why in case of ALS mutants, only a combinatorial treatment of GA and DL was able to restore the phenotypes."

In my opinion, this is not sufficient enough to warrant the obvious disparity. The authors should provide an in depth discussion to why not even a non-significant positive effect is observed alone. Especially if both supplement the NAD⁺ pathway.

Minor comments

The authors should state what normality distribution analysis was performed on their data set prior to choosing the statistical tests mentioned.

All block bar histograms should be removed and replaced with more transparent graphs, such as histograms showing the individual data points.

Point-by-point reply:

We deeply thank the reviewers for their in depth reviews and their very helpful suggestions. As you can find below, we tried hard to address every single comment appropriately. By doing so, we believe that the manuscript significantly improved and hope it is now acceptable for publication. All replies to the reviewers are marked in blue.

Reviewer #1 (Comments to the Authors (Required)):

Pal and colleagues report on the use of glycolic acid (GA) and D-lactate (DL) to restore defects in organelle trafficking and DNA damage response in cells harboring FUS and SOD1-associated ALS mutations. They use live cell imaging to record mitochondrial and lysosomal trafficking in proximal and distal segments of axons and laser irradiation to induce DNA damage responses and recruitment of FUS. They report that an iPSc-derived motor neuron (MN) line with FUS-R521C mutation has reduced motility (trafficking, displacement) of mitochondria and lysosomes in distal segments of axons.

We appreciate that the reviewer is summarizing our key findings so concisely. However, we would like to point out here that these findings are not limited solely to the FUS-R521C line. Rather, as clearly described, e.g., in the legend of Fig. 1 we have found these phenotypes and their rescue through GA & DL in all four mutant FUS lines used in this study, not just for FUS-R521C. We kindly refer to Fig. 1 in comparison to Fig. S1: Fig. 1B-E is showing pooled data, Fig. S1A-D the corresponding individual lines. As the inter-line-variability was not significant by one-way ANOVA (see legend of Fig. S1) we have pooled the data sets of all four mutant FUS lines in Fig. 1B-E, ditto for all Ctrl lines. Only the raw images in Fig. 1A are showing one representative line from each pool as specified in the legend (i.e. FUS-R521C and Ctrl1).

They also report that an ipsc-derived MN line with a TDP43-S393L mutation has reduced motility (trafficking, displacement) of mitochondria and lysosomes in proximal and distal segments of axons.

Along with the lines above, we have shown these phenotypes in two mutant TDP-43 lines, not just in the S393L line: Fig. 5B-E is showing pooled data of TDP43-S393L and G294V, Fig. S1A-D the corresponding individual lines. This is described in the respective legends. Only the raw images in Fig. 5A are showing one representative line from each pool as specified in the legend (i. e. TDP43-S393L versus Ctrl1).

This phenotype in the FUS mutant is reversed by 1 mM GA and DL cocktail (but not with single treatment of GA or DL). GA and DL did not attenuate the mitochondrial phenotype in the TDP43 mutant. In a HeLa cell line and in ipsc differentiated MNs with FUS-P525L mutations, recruitment of GFP-FUS to sites of DNA damage was improved by GA and DL. SOD1-D90A mutants showed abnormal mitochondrial elongation and membrane potential. GA and DL also restored these mitochondrial parameters.

Again, we have documented elongated mitochondria with reduced inner membrane potential in three mutant SOD1 lines, not just in the D90A line: Fig. 5B-E is showing pooled data of SOD1-D90A, A4V and R115G, Fig. S1A-D the corresponding individual lines. Please refer

to the respective figure legends. Only the raw images in Fig. 5A are showing one representative line from each pool as specified in the legend (i.e. SOD1 D90A versus Ctrl1).

This work has important strengths. The work was hypothesis driven. The group has a variety of human ipsc lines that can be used. A lot of work was performed with the directed differentiation of ipscs to MNs, the live cell imaging, and then the analysis. The figures are generally clear.

Response: We are pleased by such a great overall review.

This work has important weaknesses.

1. The manuscript as a whole is unfocused. As written, the mitochondrial theme does not fit nicely with the DNA repair theme. The authors need to establish a better logical flow.

We wish to thank the reviewer for drawing our attention to the link between distal axonal organelle perturbations and the nuclear DNA damage response (DDR). Such a “nucleo-axonal link” was actually revealed by us in Naumann et al. (2018). In this publication, we have documented that a functional DDR in the nucleus relies on the proper shuttling of FUS between the cytosol and the nucleus. FUS functions as a key player upstream in the DDR and this function relies critically on its proper posttranslational poly(ADP-ribosylation) mediated by PARP1 enzyme. Inhibition of PARP1 in wild type control cells leads to a failure of FUS recruitment to laser-induced DNA damage sites (DDS) and downstream recruitment of further DDR players as well as a reduction of nuclear FUS levels whereas cytosolic FUS aggregation increases (see Fig. 6 in Naumann et al., 2018), thereby mimicking the mutant FUS phenotype. Conversely, inhibition of the PARP1 antagonist PARG in mutant FUS cells leads to a prolonged retention time of FUS at the DDS and restores FUS recruitment to laser-induced DDS and its nuclear import, i.e. it rescues from its cytosolic aggregates.

Most intriguingly, our pharmacological manipulations with PARP1 and PARG inhibitors to study the nuclear role of FUS in DDR and its underlying shuttling between the cytosol and nucleus always had an impact on the distal axonal organelle perturbations: inhibition of FUS recruitment to DDSs in the wild type along with its cytosolic aggregation induced inevitably an inhibition of distal axonal organelle motility (lysosomes and mitochondria) and loss of mitochondrial inner membrane potential as well, even though the PARP1 inhibitor was physically absent in the distal axonal MFC compartment (detailed in Naumann et al., 2018; and Pal et al., 2018). Conversely, when we restored the nuclear FUS levels and its function at DDSs through PARG inhibitor in the FUS mutants, we always restored the distal axonal organelle phenotypes (motility, shape, inner membrane potential) as well, even though the PARG inhibitor was physically absent in the distal axonal MFC compartment (detailed in Naumann et al., 2018; and Pal et al., 2018). Finally, we found that FUS-mediated DDR in the nucleus was always functionally positioned upstream of the distal axonal organelle phenotypes, as pharmacological inhibition of mitochondrial function and motility directly at the distal axon compartment had no impact on the DDR.

Since these findings in Naumann et al. (2018), we have postulated a nucleo-axonal crosstalk to account for the evidenced functional link between FUS-mediated DDR in the nucleus and distal axonal organelle trafficking/function without having uncovered its underlying molecular mechanism so far. When we discovered that GA and DL were capable of restoring the distal axonal organelle perturbations, we were wondering if the underlying rescue mechanism might be coupled to a rescue of FUS-mediated DDR in the nucleus through the

postulated nucleo-axonal crosstalk as well. We have thoroughly revised the relevant section in Introduction, Result and Discussion to provide a more detailed account on this connection for a better clarity.

2. The different themes (mitochondrial and DNA repair) are both incompletely pursued, despite having many cell lines available. For example, with the different FUS mutations, was distal axon organelle trafficking deficit a common finding? Was DNA repair recruitment abnormally common with all the FUS mutations (they had 5 mutant lines)? Can it be established that those abnormalities in mitochondrial elongation and mitochondrial membrane potential are common among the SOD1 mutants (they had 3 different mutant lines)? Thus, as presented the work appears inconclusive and spotty.

There seems to be a misunderstanding here and we hope that our first comment above has already clarified this point to some extent. As already pointed out, all four mutant lines of the FUS pool (R521C, R521L, R495X, P525L-GFP, see Table 1) exhibit very similar distal axonal phenotypes (lysosomal and mitochondrial trafficking, reduced mitochondrial inner membrane potential, mitochondrial fragmentation; compare Figure 1 with S1 and their legends). The similarity of these phenotypes across these mutant FUS line was already published by us in Naumann et al. (2018). The main aim of this study is not to reveal and characterize these common phenotypes but to demonstrate their similar rescue through GA and DL in all four mutant FUS lines. We have revised the relevant Results section accordingly to illuminate better on our already published, common FUS phenotypes and to emphasize more on the similarity of the GA&DL-rescue across all FUS lines in this study. Regarding the phenotype in DNA damage response (DDR, Fig. 2) and its rescue through GA and DL, these studies were relying on the GFP-tag of FUS for the live imaging of the FUS recruitment-dissociation events. Therefore, this assay was solely limited to the FUS-P525L-GFP line as the other FUS lines were not engineered in their endogenous loci (see Table 1). Likewise, all three mutant SOD1 lines used in this study (SOD1-D90A, A4V, R115G, see Table 1) exhibit similar phenotypes concerning axonal mitochondrial elongation and reduced inner membrane potential. These were already published by us in Günther et al. (2022) and one key finding of this study is the successful rescue in all three SOD1 lines through GA and DL (compare Fig. 5 with S1 and their legends). Again, we have revised the relevant Results section to provide better clarity.

Likewise, both mutant TDP43 lines used in this study (TDP43 S393L and G294V, see Table 1) exhibit similar axonal phenotypes (lysosomal and mitochondrial trafficking at unaltered mitochondrial inner membrane potential and shape; compare Figure 5 with S1 and their legends). These were already published by us in Kreiter et al. (2018) and one key finding of this study is the failure of GA and DL to rescue these phenotypes in both lines. Again, we have revised the relevant Results section to provide better clarity.

Collectively, we wish to thank the reviewer for her/his comment that was very helpful in clarifying these misunderstandings in our text revisions. We hope that our revised manuscript no longer appears spotty and inconclusive.

3. The introduction (and the manuscript title) has content related to DJ-1, giving the impression that the following work might have something to do with DJ-1. The authors did not make a good connection with DJ-1. GA and DL can come from sources other than DJ-1. The DJ-1 content could be minimized.

We agree with the reviewer and significantly changed the introduction and discussion part of the revised manuscript.

4. Given the number of outliers in the graphs, did the data pass tests for normality?

We presume the reviewer is referring to the top outliers, e.g., in the box plots of Fig. 1B-D. Indeed, these are indicative of the asymmetric, top-tailed distributions of our organelle tracking and shape analysis (each data point corresponds to one organelle) which the reviewer has spotted correctly. These top-tailed distributions of mean speed, track displacements and other motility parameters are very typical for mitochondria and lysosomes and were already characterized by us, e.g., in Naumann et al. (2018) and Pal et al. (2018). This is simply due to the fact that the majority of the organelle population is of relatively low motility or even stationary, whereas only a minor fraction moves at higher speed and track displacement (i.e. processively over straight tracks), thereby creating the typical top-tailed distribution with many top outliers and with the median (centre line in box plots, see legend) always positioned clearly underneath the mean (cross in box plots, see legend). Therefore, we already established in Naumann et al. (2018) that these tracking parameters (e.g., mean speed and track displacement) as well as mitochondrial elongation (e.g., in Fig. 1D) do not pass the normality test (i.e., no bell-shaped, Gaussian distribution) using, e.g., the D'Agostino-Pearson test. Therefore, in our one-way ANOVAs, we have used the Kruskal-Wallis post hoc test to account for these asymmetric distributions. Conversely, we found that the distributions for mitochondrial inner membrane potential (JC-1) pass the normality test very well (e.g., in Fig. 1E), consistent with the much more convergent median (centre line in box plots) and mean (cross in box plots) and the well-balanced outliers at the bottom and top. The same applies for the cytoplasmic mislocalization of FUS in the mutant (e.g., in Fig. 2B). Therefore, we have used the Bonferroni post hoc test instead for inner membrane potential and cytoplasmic mislocalization. We wish to thank the reviewer for having raised this point as we found that these two different post hoc tests were not accurately described in our original manuscript, we wish to apologize for this flaw. We have now carefully revised all relevant legends and the section "Statistical analyses of box and scatter dot plots" in Material and Methods to make clear which test was used for each graph and why it was used.

5. It is not made clear whether each data point in the graphs represents one axon.

Please refer to the legends of the respective figures, each data point in the box plots represents one organelle. For example, in the legend of Fig. 1, panel B-E, we state: "Box plots quantifications of various tracking and morphology parameters deduced from movies from (A) as per organelle values..." and in the revised version we have now added "... (i.e. each data point presents one individual organelle)... except of (E) showing mean values per image". Moreover, we have revised the relevant section "Organelle tracking and shape analysis of live imaging movies" in Material and Methods to make clear that each data point of the plots mostly corresponds to one analysed organelle or in few cases to averaged values per image. This is our established and published way of presenting our organelle analysis (e.g., Naumann et al., 2018; Pal et al., 2018; Kreiter et al., 2018; Günther et al., 2022). We hope we are now providing more clarity and transparency about this point.

6. The mitochondrial and lysosomal trafficking data should be normalized to total axon length. The mutants might have shorter (or longer) axons than controls.

As clearly stated throughout the manuscript and specified in Material and Methods (see revised section “Generation, gene-editing and differentiation of human iPSC cell lines to MNs in microfluidic chambers (MFCs)” herein), we have used a compartmentalized neuronal culture system, namely Zona microfluid chambers (MFCs). As common also for other compartmentalized systems (e.g., Campenot chambers), neurons are seeded at relatively high density into a proximal compartment. In the course of the further maturation, only the protruding axons, and not the dendrites, are capable of fully penetrating a microgroove barrier of microchannels to eventually sprout out from the distal channel exits (shown, e.g., in our previous publication Glaß et al., 2020). Our set-up of compartmentalized MFC cultures is well established, published and illustrated in Fig. 2A of Naumann et al. (2018), in Fig. 1, 2A and 4A of Pal et al. (2018), in Fig. 1A of Pal et al. (2021), in Fig. 4B of Günther et al. (2022) and in Fig. 2A of Kandhavivorn et al. (2023) (see revised Material and Methods, section “Generation, gene-editing and differentiation of human iPSC cell lines to MNs in microfluidic chambers (MFCs)”).

Due to the dense, filthy nature of the somadendritic proximal compartment, the densely packed axon bundle inside the microchannels and optical resolution limits, it is not possible to track a single axon from its soma throughout the entire microchannel to its growth cone beyond the distal channel exit. Therefore, we cannot normalize our organelle tracking and shape parameters to total axon length or somehow correlate these parameters to total axon length. We can only deduce bulk per-organelle statistics at defined positions, which are the distal versus the proximal microchannel readout position (~900µm from each other) without assigning an individual organelle to a specific axon within the whole bundle. In addition, at the time point of analysis (21 DIV), all axons have already fully penetrated the microchannels, as all lines (controls and mutants) need typically less than 14 DIV to reach the distal exits. Under these conditions, we found, e.g., the axonal trafficking deficits in the FUS mutants at the distal readout position highly reproducible. Therefore, at least for both the distal and proximal readout position within the microchannels (e.g., Fig. 1A), we can state that we had apparently similar axon bundles in the all mutants as compared to the Ctrl line pool. This was also evident from the total number of organelles seized by our live imaging analysis in our bulk statistics. However, we agree with the reviewer that we cannot formerly rule out the possibility that the mutant lines in this study might have an axonal growth defect located even more distally outside the microchannels, i.e., beyond the distal channel exits in the open space which was not captured by our analysis. However, such far axonal growth defects were beyond the scope of this study and would not question the main conclusions, i.e. the promising therapeutic rescue potential of GA and DL per se. Rather, in the best-case scenario they might just help to further refine the therapeutic potential of these substances.

7. The mitochondrial and lysosomal trafficking data could also be normalized to whether that axon has a target contact.

We appreciate this comment. Our established and published differentiation pipeline (e.g., Naumann et al., 2018; Pal et al., 2018) finally yields matured motoneurons (MNs) in compartmentalized MFCs. We have previously verified that our MFC cultures are quasi 100% axon-pure in the distal compartment, see, e.g., Fig. 2B and 2E in Naumann et al. (2018) or Fig. 2F in Glaß et al. (2020). Most importantly, our MNs are not co-cultured with

any other cells that could serve as target cells, e.g., myotubes. We have revised the relevant section “Generation, gene-editing and differentiation of human iPSC cell lines to MNs in microfluidic chambers (MFCs)” in Material and Methods to make this more clear. Moreover, our MNs cannot connect to themselves or any other neuronal or non-neuronal subtypes that might arise as by-products in our differentiation pipeline because the distal compartment is completely devoid of any seeded cells. The only possible connection therefore could be axon-axon junctions, which we were unable to assess in our set-up. Therefore, any protruding axon reaching into the microchannels and finally sprouting out at the distal channel end into the open distal compartment cannot connect to any potential target cells there.

Collectively, we agree with the reviewer that modelling motoneuronal diseases in vitro in composite cultures where the MNs connect distally via neuromuscular junctions to myotubes would certainly help to further understand the therapeutic potential of GA and DL as well as the underlying cellular events in a clinically more relevant set up. However, this level of sophistication is technically much more challenging and far beyond the scope of this study.

8. The scale bar size should be checked. Or what is shown is a bundle (cable) of axons rather than an individual axon. An unmyelinated axon should be $\sim 1 \mu\text{m}$ or less.

Indeed, the reviewer is right in assuming that our images are not showing individual axons but a whole bundle of these inside an MFC microchannel. These are typically composed of 5-20 axons. We have revised the relevant section “Generation, gene-editing and differentiation of human iPSC cell lines to MNs in microfluidic chambers (MFCs)” in Material and Methods as well as the relevant figure legends to make this clear. In essence, the scale bar of $10 \mu\text{m}$ is correct.

9. The use of the GA and DL at a concentration of 1 mM is very high for cell culture work. Generally, $100 \mu\text{M}$ is considered high. Was there an effect at lower concentrations. If effects were seen at only high concentration, the metabolic interpretation becomes nebulous.

The use of GA and DL at 1 mM is actually consistent with Toyada et al. (2014). These authors reported about beneficial effects of GA and DL on mitochondrial inner membrane potential and neuronal survival and motivated us to start our own experiments with the same concentration. In Chovsepian et al. (2022), the authors titrate for the beneficial effects of GA in their cultured cortical neurons from 2.5 to 20 mM.

Regarding effects of GA and DL at lower concentration, we have clearly referred in our original Results (sections “GA and DL restore axonal trafficking in FUS-ALS mutants” and “GA and DL restore FUS nuclear cytoplasmic mislocalization and recruitment to nuclear laser-irradiated DNA damage sites.”) to our titration experiments of GA and DL in the supplement (originally Fig. S2 and S3, now in the revised version Fig. S3 and S4). In these experiments we have titrated GA and DL together at equimolar concentrations from 50-10,000 μM ($=10\text{mM}$) in the axonal trafficking assay (lysotracker, Fig. S3) and DNA damage response assay (DDR, laser-induced DNA damage, Fig. S4) to deduce the EC_{50} . Both assays revealed very similar values for the EC_{50} ($\sim 0.5 \text{ mM}$ for each substance). Therefore, we performed our experiments \sim twofold above the EC_{50} concentration which appears appropriate. However, we agree with the reviewer that these are relatively high concentrations. Therefore, we have revised our Discussion accordingly to avoid overstatements.

10. Did 20 mM GA or DL cause any evidence of toxicity (Figure 2)

We did not observe any obvious detrimental effects even at 20 mM such as, e.g., cell shrinkage, detachment, blebbing, vacuolization, occurrence of debris, etc. We have added some clarifying sentences about this point to the relevant Material and Methods section “Treatments and inhibitors” in the revised manuscript.

11. In the work presented, what defined the cells as differentiated MNs? For example, the cells shown in Figure 2E and Figure 3B, do not look much like differentiated human MNs. The authors should prepare a supplemental figure defining their human ipsc-derived MN system.

Figure 2E, 3B and the new supplemental Fig. S6A are showing close-ups of individual motoneuronal somata because these are the relevant areas where the laser irradiations and FUS mislocalizations occurred. However, even in these close-ups, a few protruding and surrounding neurites are readily visible as dark blue, “filamentous” structures (particularly in Fig. S6A). This dim neuritic appearance is simply due to nature of the FUS localization mainly in the somata (i.e., in the nucleus and cytosol there) whereas the neurites are relatively much lower in their FUS levels. Because our image galleries are showing **STAINLESS** live images of FUS-GFP only in the FIJI look-up table (LUT) “Green Fire Blue” (see legends to Fig. 3B, S6A), low neuritic FUS-GFP levels show up just barely in dim dark blue. Therefore, live FUS-GFP images are suboptimal in order to reveal neuritic structures by IF microscopy, but that is obviously not the purpose of these figure preparations here.

The analysis in our study was done at 21 DIV. We already characterized the time course of motoneuronal differentiation of all used cell lines previously, including electrophysiology recordings showing firing of repetitive actions potentials and so on (e.g., Naumann et al., 2018, Reinhardt et al., 2013, Naujock et al., 2016), including the occurrence of common neuronal markers (i.e. β 3-tubulin, MAP2) and specific spinal motoneuronal markers (i.e. SMI32, ChAt, Islet, HB9) (see Fig. 1A-F in Naumann et al, 2018; Fig. 5A-F in Reinhardt et al., 2013; and Fig. 1A-C in Naujock et al., 2016). Moreover, our obtained MNs display electrophysiological features indicative of successful differentiation and maturation (see Fig. 1G-P in Naumann et al., 2018; and Fig. 1D-H, 2, 3, 4 in Naujock et al., 2016). Finally, the differentiation pipeline from our iPSC-derived neuronal precursor cells (NPCs) to mature MNs is illustrated in Fig. 1A of Naumann et al. (2018), in Fig. 1A of Kreiter et al. (2018), in Fig. 4A of Günther et al. (2022) and in Fig. 1A of Kandhavivorn et al. (2023). Collectively, our motoneuronal differentiation protocol was already thoroughly described and published, characterized and illustrated throughout our previous publications; thus, we wanted to avoid the “re-publication” of established findings. We have now revised section “Generation, gene-editing and differentiation of human iPSC cell lines to MNs in microfluidic chambers (MFCs)” in Material and Methods to make this more clear.

12. The authors should label in the figure panels and graphs whether the results are from HeLa cells or MNs.

We did so in our revised figures.

13. In Figure 2C and E, why can't the endogenous non-recruited FUS be seen in the nucleus?

First of all, we would like to remind that these images are only showing FUS-GFP in living cells, i.e., these are stainless IF images shown in the LUT “Green Fire Blue”, as indicated in the corresponding legends. Therefore, all dark blue areas inside the nucleus are NOT showing a HOECHST staining or alike. Rather, they are indicating low FUS-GFP levels whereas high local concentrations of the recruited FUS-GFP at the laser cut are shown in bright green. Maybe the low FUS-GFP level in dark blue surrounding the laser cut site is actually the signal that the reviewer was missing as “endogenous non-recruited FUS” in the nucleus. Regarding the endogenous, i.e. non-tagged, FUS, this molecular population remains invisible in these stainless IF images. These might be addressed by IF stainings, however the kinetics will vary significantly due to time of fixation prior to staining, and only few defined timepoints can be analysed.

Reviewer #2 (Comments to the Authors (Required)):

The authors describe the effects on several cellular parameters of two molecules that are putatively produced by the enzymatic activity of DJ-1.

The cellular models used are well constructed and represent a wide spectrum of ALS mutations, this aspect is certainly a strong point of the work itself. The quantity of experiments and the quality of these, well characterize the effects of glycolate and D-lactate on axonal mitochondrial transport, on the polarization of the mitochondrial membrane, on the aggregation state of the FUS, etc.

Response: We deeply appreciate the positive overall impression of our manuscript.

In my opinion, however, the work describes, without giving a mechanistic motivation (for example why are the two molecules effective only in combination?), the action of two molecules that can be produced also through pathways that do not involve the activity of DJ-1. Indeed, the glyoxylase or deglycase activities of DJ-1 are currently widely discussed, and only a few papers, mainly conducted in in vitro models, have suggested such activities among the numerous pleiotropic roles described for DJ-1. Among other things, the putative glyoxylase activity of DJ-1 produces a mixture of D-lactate and L-lactate, and the authors never used both enantiomers.

I therefore find it inappropriate to mention DJ-1 in the title as a major player in the production of such compounds.

At the very least, the authors should have explored the expression levels of DJ-1 in the cell lines used or modified its expression.

We appreciate the reviewer's concern about our statements of DJ-1 and its putative key role in the mode of action of GA and DL. Therefore, we have carefully revised the relevant parts to tone the role of DJ-1 down to one plausible underlying mechanism amongst others. Along these lines, we have added two new supplemental figures, i.e., S2 and S5, showing additional control experiments with L-lactate (LL) alone or in combination with GA in direct comparison to D-lactate (DL) alone and the combination of GA and DL to clarify the role of both lactic enantiomers. In essence, LL did not show any effect neither when administered alone nor in combination with GA in our axonal trafficking (Fig. S2) and laser-induced DNA damage response (DDR) assay (Fig. S5). In our revised Discussion about feasible rescue mechanisms, we take these additional enantiomeric control data into consideration.

Nevertheless, we prefer to also still ~~discussed~~ the data from Toyoda et al. on DJ-1 knockouts and the rescue of mitochondrial depolarization by GA/DL in those as -t ~~These~~ data give some evidence that GA/DL can rescue loss of DJ-1 function.

In short, I find a weak connection between the experimental structure of the work and the title, which certainly gives different expectations, for example the study of the role of DJ-1 in the ALS patients-derived cell models used.

AS written above, we tried hard to rephrase the whole manuscript appropriately.

Reviewer #3 (Comments to the Authors (Required)):

This paper by Pal and colleagues, studies disease mechanisms in iPSC-derived motor neurons from mutant SOD1 and FUS patients. Glycolic acid and lactic acid are shown to counteract some of the phenotypes seen in vitro, in particular deficient axonal trafficking and DNA damage response. The manuscript is of interest to the field, confirms the neuroprotective effect of GA and DA (as previously shown on dopaminergic neurons) and provides avenues for further research.

My comments:

- Abstract: the last paragraph of the abstract is too optimistic. GA and DA have maybe shown positive effects in vitro, but the way to a drug candidate is long. In addition, the concentrations needed are quite high.

We tried hard to rephrase the whole manuscript appropriately. In addition, we have added a significant limitation section to the revised Discussion.

- Introduction: the introduction is very long, could be more concise and focused on the research question.

We streamlined the introduction.

- introduction: references for these statements are missing "DJ-1 overexpression protects dopaminergic neurons against PD, whereas DJ-1 deficiency leads to profound loss of dopaminergic neurons".

We added this reference.

- Statistics: one-way ANOVA tests are used when comparing multiple groups, but the p-value shown on the figures is always the comparison with the control I suppose. For mutant conditions with and without treatment, the p-value for this comparison is most relevant and should be displayed as well. This comment is applicable to multiple figures.

We appreciate the point the reviewer is raising and have carefully revised the highlighted significances of our ANOVA tests throughout all figures of question. All box and scatter dot plots (e.g., Fig. 1B, 2B, etc.) display now brackets on top of the data to indicate the significances of two kinds of pairwise comparisons:

1. For a given cell line (e.g., mutant FUS), whether a treatment (e.g., GA & DL) as compared to its untreated/Mock condition led to a significant change (asterisks) or not (ns).
2. For a mutant, untreated (Mock) cell line, whether this line had a natural, significant phenotype (asterisks) as compared to Mock Ctrl or not (ns).

Please note that often the majority of all pairwise comparisons was not significant (e.g., within the proximal parts of Fig. 1B-E). In order to avoid a too messy appearance we did not add significance indicators for these in the graphs. Instead, we have included the comment "...all other pairwise comparisons were not significantly different" in the corresponding legends.

- Results figure 2: the mislocalization/cytoplasmic aggregates of FUS are studied in HeLa cells. It would be of interest to add experiments in iPSC-derived motor neurons as well.

We have added a new supplemental Fig. S6 showing cytoplasmic mislocalization of FUS in iPSC-derived motor neurons and its successful rescue through GA and DL treatment, thereby validating our findings in the HeLa cell model (Fig. 2).

- Figure 4: DJ-1 products GA and DL restore nuclear phenotypes by restoring NAD metabolism.

The rescue by NAR does not allow to make the claim that the rescue of GA and DL is by restoring NAD metabolism. It can at most suggest it.

We agree with the reviewer and rephrased this section.

- Figure 5: GA and DL together rescue axonal organelle phenotypes in ALS-SOD1 but not in TDP-43 mutants.

Only transport data are shown, so the title should maybe be changed.

We have chosen the generic term “axonal organelle phenotypes” instead of, e.g., “axonal trafficking phenotypes” because Fig.5 comprises both, an organelle shape/functional analysis (i.e., the hyper-elongation of mitochondria along with reduced inner membrane potential in the mutant SOD1 line pool, Fig. 5D, E) as well as a trafficking (i.e. motility) analysis (i.e., track displacement and mean speed in the mutant TDP43 line pool, Fig. 5B, C). The term “axonal organelle phenotypes” simply meant to cover both types of phenotype. However, for the sake of better clarity, we have now revised this title to: “Figure 5: GA and DL together rescue mitochondrial hyper-elongation and inner membrane potential in ALS-SOD1 axons but not the mitochondrial and lysosomal trafficking defects in TDP-43 mutants.”

Reviewer #4 (Comments to the Authors (Required)):

The authors have focussed on the glyoxylase DJ-1 and the downstream products glycolic acid (GA) and D-lactic acid (DL). The authors have supplemented MNs from patient ALS models specifically FUS, SOD1 and TDP43 with GA and DL and measured mitochondrial and lysosomal axonal trafficking, mitochondrial membrane potential, elongation and protein aggregation. Treatment with GA and DL restored axonal trafficking deficits of mitochondria, restored membrane potential and elongation as well as and DNA damage recruitment. The authors have showed that GA and DL combined restore the defects observed in FUS models, partially in SOD1 models but not in TDP43 models.

Overall, certainly in terms of FUS, this is a well thought out approach with robust data sets presented in a transparent manner. The data is clear and the discussion points are valid.

In my opinion the SOD1 data is less clear and somewhat muddles the message. However, this could be clarified with further discussion. The issue arises around the "effect" on axonal trafficking of GA and DL in the SOD1 models. In the results section the author's state

"GA and DL restore axonal trafficking in SOD1- but not TDP43-ALS mutant"

In the discussion the author's state

"We show that combinatorial treatment with GA and DL restored axonal trafficking deficits of mitochondria and lysosomes in FUS- and SOD1-ALS"

In my opinion GA and DL have no effect on mitochondrial or lysosomal trafficking in SOD1 models. They have a clear effect on mitochondrial membrane potential and shape (response to JC-1 and elongation), but no effect on track displacement and mean speed, probably because the SOD1 MNs don't seem to have defects in the latter two?

We wish to thank the reviewer for this very detailed review of our dataset and we apologize ~~forte~~ not being clear enough. Indeed, the SOD1 mutants have no phenotype in their organelle motility (i.e., mean speed and track displacement of lysosomes and mitochondria, Fig. 5B, C) but instead only in the shape of their mitochondria (i.e., hyper-elongation, Fig. 5D) and inner membrane potential (JC-1, Fig. 5E). We went carefully through the manuscript and have now rephrased all statements about the SOD1 phenotypes and their rescue through GA and DL. For example, in Results we have replaced the original subtitle "GA and DL restore axonal trafficking in SOD1- but not TDP43-ALS mutants." by "GA and DL restore axonal organelle phenotypes in SOD1- but not in TDP43-ALS mutants." The nature of these "axonal organelle phenotypes" is then further specified in the following revised text.

Moreover, SOD1 mitochondria are actually more elongated compared to controls compared to FUS which are less elongated. However, GA and DL correct both? So there needs to be careful re-wording of the manuscript and clarity on the precise modes of mechanisms.

We thank the reviewer for this point. We rephrased this part, please also see comments to other reviewers on this.

One major aspect of the paper overall is that GA and DL alone have no effect at all, which is very surprising. The authors state

"It remains, however, open, why in case of ALS mutants, only a combinatorial treatment of GA and DL was able to restore the phenotypes."

In my opinion, this is not sufficient enough to warrant the obvious disparity. The authors should provide an in depth discussion to why not even a non-significant positive effect is observed alone. Especially if both supplement the NAD⁺ pathway.

We thank the reviewer for raising this point. We tried hard to make this much clearer in the revised version, including a limitation section in Discussion.

Minor comments

The authors should state what normality distribution analysis was performed on their data set prior to choosing the statistical tests mentioned.

We thank the reviewer for raising this point and like to refer also to Reviewer 1 point 4. We presume the reviewer is referring to the top outliers, e.g., in the box plots of Fig. 1B-D. Indeed, these are indicative of the asymmetric, top-tailed distributions of our organelle tracking and shape analysis (each data point corresponds to one organelle) which the reviewer has spotted correctly. These top-tailed distributions of mean speed, track displacements and other motility parameters are very typical for mitochondria and lysosomes and were already characterized by us, e.g., in Naumann et al. (2018) and Pal et al. (2018). This is simply due to the fact that the majority of the organelle population is of relatively low motility or even stationary, whereas only a minor fraction moves at higher speed and track displacement (i.e. processively over straight tracks), thereby creating the typical top-tailed distribution with many top outliers and with the median (centre line in box plots, see legend) always positioned clearly underneath the mean (cross in box plots, see legend). Therefore, we already established in Naumann et al. (2018) that these tracking parameters (e.g., mean speed and track displacement) as well as mitochondrial elongation (e.g., in Fig. 1D) do not pass the normality test (i.e., no bell-shaped, Gaussian distribution) using, e.g., the D'Agostino-Pearson test. Therefore, in our one-way ANOVAs, we have used the Kruskal-Wallis post hoc test to account for these asymmetric distributions. Conversely, we found that the distributions for mitochondrial inner membrane potential (JC-1) pass the normality test very well (e.g., in Fig. 1E), consistent with the much more convergent median (centre line in box plots) and mean (cross in box plots) and the well-balanced outliers at the bottom and top. The same applies for the cytoplasmic mislocalization of FUS in the mutant (e.g., in Fig. 2B). Therefore, we have used the Bonferroni post hoc test instead for inner membrane potential and cytoplasmic mislocalization. We wish to thank the reviewer for having raised this point as we found that these two different post hoc tests were not accurately described in our original manuscript, we wish to apologize for this flaw. We have now carefully revised all relevant legends and the section "Statistical analyses of box and scatter dot plots" in Material and Methods to make clear which test was used for each graph and why it was used.

All block bar histograms should be removed and replaced with more transparent graphs, such as histograms showing the individual data points.

We appreciate the reviewer's request for better transparency and have replaced all bar graphs by scatter dot plots (revised Fig. 2B, 3D, 4B, S5B, S6B). The resultant scatter dot clouds

comprise typically 30 data points, each representing a mean value per image as specified in the corresponding legends (e.g., of Fig. 2B) and in Material and Methods, section “Image/movie quantification”.

Regarding the remaining box plots (e.g., Fig. 1B), these are showing per-organelle values, i.e. each data point corresponds to one organelle (as specified in the corresponding legends and in Material and Methods, section “Organelle tracking and shape analysis of live imaging movies”). Already from the large number of outliers outside the whiskers (which cover 99% of the data, as specified in the legends), one can already deduce a very high number of organelles (sample size) underlying each box plot, actually tens of thousands for the line pools in, e.g., Fig. 1B. Therefore, we would prefer to keep these graphs as box plots and not to convert them to scatter dot plots as the resultant data dot clouds would become too dense and messy. This is particularly the case for Fig. S1 showing many narrow box plots in parallel of all individual lines which are already hard to discern in the current state and would become even more incomprehensible if converted to scatter dot plots. In these cases, the simplifying abstraction enabled by the box plots provides more clarity rather than showing each individual data point in our opinion. Moreover, this is consistent with our established and proven way of presenting our organelle tracking analysis throughout our previous publications (e.g., Naumann et al., 2018; Pal et al., 2021; Kreiter et al., 2018; Günther et al., 2022), which thus would allow for a better comparison to those.

April 25, 2024

RE: Life Science Alliance Manuscript #LSA-2023-02535-TR

Prof. Andreas Hermann
University of Rostock
Gehlsheimer Str. 20
Rostock 18147
Germany

Dear Dr. Hermann,

Thank you for submitting your revised manuscript entitled "Glycolate and D-lactate - putative products of DJ1 - restore neurodegeneration in FUS- and SOD1-ALS". We would be happy to publish your paper in Life Science Alliance pending final revisions necessary to meet our formatting guidelines.

- please be sure that the authorship listing and order is correct
- please add Keywords and a Summary Blurb/Alternate Abstract to our system
- please add the Twitter handle of your host institute/organization as well as your own or/and one of the authors in our system
- please note that the titles in the system and manuscript file must match
- please be sure that all authors are mentioned in the author contribution section in your manuscript file
- please mention panel B in the figure legend for Figure S6
- please correct the few first references marked Invalid in the Reference list
- please use the [10 author names et al.] format in your references (i.e., limit the author names to the first 10)
- please add callouts for Figures S2C; S3B; S4B; S6A and S7B to your main manuscript text

A. FINAL FILES:

B. MANUSCRIPT ORGANIZATION AND FORMATTING:

Thank you for your attention to these final processing requirements. Please revise and format the manuscript and upload materials within 4 days.

Sincerely,

May 7, 2024

RE: Life Science Alliance Manuscript #LSA-2023-02535-TRR

Prof. Andreas Hermann
University of Rostock
Gehlsheimer Str. 20
Rostock 18147
Germany

Dear Dr. Hermann,

Thank you for submitting your Research Article entitled "Glycolic acid and D-lactate-putative products of DJ-1-restore neurodegeneration in FUS and SOD1-ALS". It is a pleasure to let you know that your manuscript is now accepted for publication in Life Science Alliance. Congratulations on this interesting work.

DISTRIBUTION OF MATERIALS:

Again, congratulations on a very nice paper. I hope you found the review process to be constructive and are pleased with how the manuscript was handled editorially. We look forward to future exciting submissions from your lab.

Sincerely,
